



# Electricity savings and greenhouse gas emission reductions from global phase-down of hydrofluorocarbons

Pallav Purohit[1], Lena Höglund-Isaksson[1], John Dulac[2], Nihar Shah[3], Max Wei[3], Peter Rafaj[1], and Wolfgang Schöpp[1]

[1]Air Quality and Greenhouse Gases (AIR) Program, International Institute for Applied Systems Analysis (IIASA), Schlossplatz 1, A-2361, Laxenburg, Austria
[2]International Energy Agency (IEA), 9, rue de la Federation, 75015 Paris, France
[3]Energy Technology Area, Lawrence Berkeley National Laboratory (LBNL), 1 Cyclotron Road, Berkeley, CA 94720, USA

*Correspondence to*: Pallav Purohit (purohit@iiasa.ac.at)

**Abstract.** Hydrofluorocarbons (HFCs) are widely used as cooling agents in refrigeration and air conditioning, as solvents in industrial processes, as fire extinguishing agents, for foam blowing and as aerosol propellants. They have been the primary substitutes for ozone-depleting substances regulated under the Montreal Protocol (MP). However, HFCs are potent greenhouse gases (GHGs) and as such subject to global phase-down under the Kigali Amendment (KA) to the MP. In this study, we develop a range of long-term scenarios for HFC emissions under varying degrees of stringency in climate policy and assess

co-benefits in the form of electricity savings and associated reductions in GHG and air pollutant emissions. Due to technical opportunities to improve energy efficiency in cooling technologies during the phase-down of HFCs, there exist potentials for significant electricity savings under a well-managed phase-down of HFCs. Our results show that annual pre-KA baseline emissions of HFCs are expected to increase from almost 0.5 to about 4.3 Gt $CO_2$eq between 2005 and 2050 and reach between 6.2 and 6.8 Gt $CO_2$eq in 2100. The growth is driven by a strong increase in demand for refrigeration and air conditioning

services, which in turn is driven by an expected increase in per capita wealth in developing countries and a warmer future climate. We estimate that full compliance with KA means cumulative global HFC emissions that are 87% lower than in the pre-KA baseline between 2018 and 2100. Also, the opportunity to simultaneously improve energy efficiency in stationary cooling technologies during such a transition could bring about additional climate benefits of about the same magnitude as that attributed to the phase-down of HFCs. If technical energy efficiency improvements are fully implemented, the resulting

electricity savings could exceed a fifth of future global electricity consumption. Together with a HFC phase-down, this means preventing between 390 and 640 Gt $CO_2$ equivalent of GHG emissions between 2018 and 2100, thereby making a significant contribution towards keeping the global temperature rise below 2°C. Reduced electricity consumption also means lower air pollution emissions in the power sector, estimated at about 10% for $SO_2$, 16% for $NO_x$ and 9% for $PM_{2.5}$ emissions, compared with a pre-KA baseline.





## 1 Introduction

Hydrofluorocarbons (HFCs) are widely used as cooling agents in refrigeration and air conditioning, as solvents in certain industrial processes, as fire extinguishing agents, for foam blowing and as aerosol propellants. As well, HFC-23 is generated as a by-product of chlorodifluoromethane (HCFC-22) production for feedstock and emissive use. HFC emissions have increased significantly in recent years in response to increased demand for cooling services and the phase-out of ozone-

depleting substances (ODS) under the Montreal Protocol (MP) (UNEP, 2007; Velders et al., 2009, 2012, 2015; Gschrey et al., 2011; Fang et al., 2016, 2018; Purohit and Höglund-Isaksson, 2017). HFCs are potent greenhouse gases (GHGs) with a global warming potential (GWP) up to 12400 times that of $CO_2$ per mass unit (IPCC, 2013). As users phase out chlorofluorocarbons (CFCs) and hydrochlorofluorocarbons (HCFCs) under the MP, they have often made choices between high-GWP HFC alternatives and alternatives that are more climate-friendly, e.g., hydrocarbons, ammonia, pressurized carbon dioxide and

unsaturated HFCs (HFOs). In particular countries subject to Article 5 under the MP (i.e., developing countries) now have the opportunity to leapfrog from the current use of HCFCs and HFCs to alternative technologies with low global warming potential (low-GWP) that are often also more energy efficient (UNEP, 2016a).

The Kigali Amendment (KA) to the MP agreed in October 2016 and which entered into force on January 1, 2019, is a global agreement to phase down and close to eliminate the consumption of HFCs by 2050 (UNEP, 2016b). Under the KA agreement,

countries have been attributed to four different party groups, in which each is subject to an emission reduction schedule outlining target reduction over the next three decades. While previous MP agreements have resulted in improvements in the design and energy performance of equipment (IPCC/TEAP, 2005), the KA is the first time that maintaining and/or enhancing the energy efficiency of equipment is explicitly included as a goal (EIA, 2016). Hence, the environmental impact of a transition away from HFCs is not only associated with the radiative properties and lifetime of the cooling agents, but also with the carbon

dioxide ($CO_2$), methane ($CH_4$), and associated air-pollution emissions of the energy used to power the cooling equipment over its entire lifetime. The switch to low-GWP cooling technology accordingly offers an opportunity to redesign equipment and improve its energy efficiency (UNEP/CCAC, 2016). Much due to a lack of detailed estimations at the sector/technology- and HFC species levels, there is currently limited understanding of the potential future impacts of the KA on global warming and possible co-benefits from savings in electricity (Shah et al., 2019). This study is, as far as we are aware, the first attempt to try

to quantify the overall effects of the KA on both greenhouse gas and air pollutant emissions. Similarly, there is a need to better understand the implications of going beyond the KA targets and aiming at a close to complete phase-out of HFC emissions globally at an earlier point in time than required under the KA. Addressing these knowledge gaps is the purpose of this study. The Greenhouse gas - Air pollution Interactions and Synergies (GAINS) model developed by the International Institute for Applied Systems Analysis (IIASA) has previously been used to produce detailed future scenarios for HFC emissions to 2050

(Höglund-Isaksson et al., 2017; Purohit and Höglund-Isaksson, 2017), which have fed into climate models to assess potential impacts on global warming (e.g., UNEP/CCAC, 2018; Rogelj et al., 2018; UNEP, 2017; Gambhir et al., 2017). This study extends on previous work by producing long-term scenarios of HFC emissions to the year 2100 under varying degrees of



stringency in climate policy, and by assessing potential co-benefits in the form of savings in electricity and associated reductions in greenhouse gas and air pollution emissions.

The paper is set out as follows: Section 2 presents the methodology used to generate baseline and alternative scenarios for HFC emissions and for estimating potentials for electricity savings in the cooling sector. Section 3 presents the low-GWP options considered as replacements for the use of high-GWP HFCs in the GAINS model. Section 4 presents results while Section 5 concludes the key findings of the study.

## 2 Methodology

### 2.1 Baseline scenarios

For the purpose of this study, baseline scenarios for global HFC emissions have been developed under the assumption that the KA is not implemented. Although pre-KA baseline scenarios may be seen as outdated and therefore uninteresting given that the KA has already entered into force[1], it is still necessary to first generate baselines as consistent bases for the construction of future emission reduction scenarios. The demand for cooling is here expressed in terms of equivalent mass units of HFCs

consumed. The starting point is the current consumption of HFCs by species and sector as reported by countries to the United Nations Framework Convention on Climate Change (UNFCCC) or derived in the GAINS model using a consistent methodology (Purohit and Höglund-Isaksson, 2017). To the extent that alternative technologies are already adopted due to existing national and regional regulations (see: Section S1 of the Supplementary Information (SI)), impacts are reflected in both historical HFC consumption levels and in future baseline scenarios. Future demand for HFCs in a pre-KA setting is

projected using population, macroeconomic variables (GDP and value added from industry and services) and cooling degree days (CDDs) as drivers and under the assumption that the use of HFCs for cooling continues into the future. The pre-KA baseline scenarios provide a primary point of reference for evaluating the need for, and impact of, alternative technologies. Hence, the mitigation scenarios developed here assume the same demand for cooling services as in the respective baselines, but with the consumption of high-GWP HFCs replaced by alternative low-GWP technologies. The choice and order of adoption

of technologies in a given sector are determined by marginal abatement cost curves (MACC) estimated on the basis of baseline HFC consumption (Höglund-Isaksson et al., 2017). For descriptions of key drivers at the sectoral level, source-specific emission factors and implemented control policies, see the supplementary material of Purohit and Höglund-Isaksson (2017).

The baseline scenarios improve upon those presented in Purohit and Höglund-Isaksson (2017) and Höglund-Isaksson et al., (2017) not only by extending the scenarios to 2100, but also by making use of the information on historical HFC consumption

by sector and HFC species that has recently become available at increasingly greater detail from the National reporting to the UNFCCC. The principal information sources used to estimate historical HFC consumption and emissions are: 1) robust

---

[1] Ninety-three signatories have ratified the Kigali Amendment to the Montreal Protocol on phasing down HFCs worldwide until 24th February 2020 (UN, 2020).





historical HFC consumption data by sector (2005, 2010 and 2015) for developed countries derived from their UNFCCC National Inventory Submissions (UNFCCC, 2017); 2) historical HFC consumption data for China and India and with some additional information for other developing countries from various national and international sources[2]; 3) data on historical

HCFC consumption from UNEP (2017), part of which has been replaced by HFCs; and 4) assumed effective control of HFC-23 ($CHF_3$) emissions from the manufacture of HCFC-22 ($CHClF_2$) in China (Simmonds et al., 2018; UNEP, 2018) and India (GoI, 2016; Say et al., 2019). From these compiled datasets, historical HFC consumption is derived for 174 countries/regions and for 14 separate source sectors (including aerosols, commercial refrigeration, domestic refrigerators, fire extinguishers, ground source heat pumps, HCFC-22 production for emissive and feedstock applications, one component and other foams,

industrial refrigeration, mobile air-conditioning, solvents, stationary air-conditioning (including commercial and residential), and transport refrigeration), and 13 different HFC species (HFC-23, HFC-32, HFC-125, HFC-134, HFC-134a, HFC-143, HFC-143a, HFC-152a, HFC-245fa, HFC-365mfc, HFC-43-10mee, HFC-227ea, HFC-236fa). Blends of HFCs have been decomposed and attributed to respective HFC species. In this study, we apply IPCC AR5 global warming potentials (GWPs) over 100 years without climate–carbon feedback effects when converting the warming potential for different HFC species to

$CO_2eq$ units (IPCC, 2013). Moreover, the commercial refrigeration and air-conditioning sectors are subdivided into small and large systems to allow for adoption of different low-GWP alternatives for small and large units in mitigation scenarios. The same level of detail at the country-, sector- and HFC species levels as for historical emissions, are maintained in the construction of future emissions scenarios.

For the development of the baseline scenarios in the timeframe to 2040, we use the existing model setup in GAINS, which for

global scenarios uses drivers consistent with macroeconomic and energy sector projections from the IEA World Energy Outlook 2017 (IEA-WEO, 2017)[3]. For stationary air-conditioning, the global stock of air conditioners in buildings modelled in GAINS were adjusted to approximate the stocks estimated by the IEA (2018). The extension in demand for cooling services between 2040 and 2100, expressed here in tons of HFC consumed, has been generated in consistency with the growth in population and macroeconomic indicators of the third Shared Socioeconomic Pathway (SSP3) (IIASA, 2017)[4] and the

expected future increase in regional CDDs received from IEA (2018). The reason for selecting the SSP3 scenario as the primary driver for the extension to 2100 is that for the period 2005 to 2040 it comes the closest to the IEA-WEO (2017) in terms of growth in global population and GDP levels (see: Figure S1 of the SI). The SSP3 is, however, a relatively pessimistic future scenario with the highest growth in global population and one of the lowest GDP growth rates among all SSP scenarios. We have therefore prepared alternative projections to 2100 using the more optimistic SSP1 scenario as a sensitivity case. The

---

[2] Including Climate & Clean Air Coalition (CCAC), UNEP Ozone Secretariat, non-Annex-I parties national communication to the UNFCCC, etc.

[3] GAINS relies on import of externally produced macroeconomic and energy sector projections. In this case, the range of energy sector scenarios produced for the IEA-WEO 2017 was used.

[4] The SSPs are based on five narratives describing the alternative socio-economic developments "sustainable development" (SSP1), "middle-of-the-road development" (SSP2), "regional rivalry" (SSP3), "inequality" (SSP4), and "fossil-fueled development" (SSP5) (Riahi et al., 2017).





difference in HFC emission projections compared to the SSP3, turned out to be minimal. Since the mitigation potential relative the baseline is similar for different SSP scenarios, this kind of sensitivity analysis only brings added value in a baseline setting (addressing a variation in baseline demand for cooling services). The SSP1 scenario is therefore not analysed in a mitigation context.

    Exposure to health risks due to extreme temperatures have been growing worldwide (Mueller et al., 2016; Pal and Eltahir,

2016; Mora et al., 2017; Russo et al., 2017) and a significant number of heat related deaths are reported annually during the summer months in both the northern and southern hemispheres, particularly among the elderly, the poor, and in densely populated cities (Mastrucci et al., 2019). Global heat stress is projected to increase in a 1.5°C warmer world (IPCC, 2018). Compared to a 1961–1990 level, climate change could by 2030 be responsible for additional annual deaths of 38,000 people from heat stress, particularly among the elderly (WHO, 2014). Each 1°C increase could reduce work productivity by 1-3% for

people working outdoors or without air conditioning, typically the poorer segments of the workforce (Park et al., 2015). The increased use of air conditioning enhances resilience to heat stress (Petkova et al., 2017). However, due to its high cost, air conditioning is considered a luxury, and only 8% of the 2.8 billion people living in the world's hottest regions possess an air conditioning unit today (IEA, 2018). In addition, almost one billion people lack access to electricity (IEA, 2019) and at least one billion live in slum conditions (World Bank, 2019), both of which make access to space cooling challenging. Cooling

contributes significantly to peak load and is therefore an important consideration when deciding on capacity of electricity networks (Shah et al., 2015). The lack of access to essential indoor cooling is a major equity issue and is increasingly seen as a dimension of energy poverty and wellbeing that demands attention from policymakers. Therefore, in parallel with the SSP3 baseline scenario and drawing on previous work by IEA (IEA, 2018), a *Cooling for All*[5] scenario has been developed, which is an alternate baseline scenario that focuses on how we embed growing cooling demands that can reach everyone within a

clean energy transition, and in turn, support faster progress to achieve the goals of the KA. In this alternate baseline scenario, we do not model demand for cooling services in the residential sector only as a function of population, macroeconomic drivers, and changes in CDDs, but assume in addition that in countries/regions with average climates exceeding 1000 CDDs[6], the uptake of residential air conditioners is at least one per household by 2050 (and takes place irrespective of income constraints). Similarly, the uptake of domestic refrigerators in the *Cooling for All* scenario is assumed to be at least one per household by

2050 irrespective of income constraints.

    Energy efficient buildings, shading, cool/green roofs etc. could substantially reduce HFC and electricity consumption in residential and commercial buildings (Goetzler et al., 2016). However, in this study we have not considered such options,

---

[5] The *Cooling for All* initiative (IEA, 2018) focuses on how we provide sustainable access to cooling within a clean energy transition, and in turn, support faster progress to achieve the goals of the Kigali Amendment to the Montreal Protocol, agreed on in Rwanda in 2016.

[6] For regions with CDD<1000 it is assumed that the households will use other cooling appliances (e.g. fan, evaporative coolers, etc.) if they cannot afford room air-conditioner. By 2050, approximately 183 million households (or nearly 1 billion people) in hot countries will have at least one air-conditioner in the *Cooling for All* scenario as compared to the SSP3 baseline scenario.



partly due to a lack of detailed information about their potential at the country level and partly due to the focus of this study on direct replacement of current and future use of HFCs with alternative substances.

Effective cooling is essential to preserve food and medicine (Peters, 2018). The increased demand for cooling to preserve food in a warmer world, including the associated increase in electricity consumption, are considered in the baseline scenarios for emissions developed here. Extended refrigeration of food would also mean reduced food losses, which apart from having important implications for meeting nutritional needs, would also contribute to reduced greenhouse gases from food production and better use of the 23–24% of global cropland and fertilizers currently used to produce food that is eventually lost (Kummu

et al., 2012; Hiç et al., 2016). Hence, reducing global food supply chain losses have several important secondary benefits including conservation of energy and other resources (Kummu et al., 2012) as these are freed up to be converted into other productive activities (Ingram, 2011; Beddington et al., 2012; Kummu et al., 2012; Hiç et al., 2016; Lamb et al., 2016). Due to a lack of detailed information on impacts on food supply chains, such secondary benefits from extended use of industrial and commercial refrigeration and refrigerated transport are not considered in this study.

**2.2 HFC reduction scenarios**

We develop alternative HFC reduction scenarios in consistency with the demand for cooling modelled in the pre-KA baseline scenarios described in Section 2.1. The key contribution of this task is not to determine the reduction levels in HFC consumption (as these are already pre-determined by the regional targets of the KA and by the almost complete reductions possible under MTFR), but to investigate the content of the HFC phase-down in terms of to what extent the various alternative

technologies identified are picked up in the different sectors and regions. This is important as it is only by understanding the content of the low-GWP technology uptake that we can get an idea of the expected degree of employment of different technologies and their respective contributions to electricity savings and reductions in GHGs and air pollution, which tend to differ by region, sector and technology (Höglund-Isaksson et al., 2017).

The order of technology uptake to meet the KA targets is determined by the marginal abatement cost curves, which for a given

technology and sector are defined using country-, sector-, and year- specific information (Höglund Isaksson et al., 2016; 2017). For example, the variation in unit costs reflects variations between countries and over time in electricity prices and labor costs. For this study, we have used marginal abatement cost curves (MACC) to simulate technology uptake every five years until 2050 and assume the relative employment of technology in 2050 to remain constant until 2100 at the country- and sector-level. Given the high uncertainty about future technology developments, we find that it does not make much sense to model

individual technology uptake in greater detail than this in the period post 2050. To model the sector technology uptake required to meet the KA, we have produced emission scenarios with cost curves including all HFC source sectors, i.e., in addition to cooling, we also include HFC emissions from aerosols, foams, industrial processes and other sources. This is necessary because the relative contribution of each sector towards the predetermined regional reduction targets (see: Section S2 of the SI) can only be determined when all HFC sectors are included in the analysis. Note that for given technology options, potential effects

of future technological development on costs, HFC removal efficiency and energy efficiency have not been considered here.





As the removal of HFCs is close to complete with existing technology and the sole purpose of cost estimates is to determine the order of technology uptake, inclusion of technological development effects will not affect conclusions regarding the HFC phase-down. However, not considering potential technological development effects on future energy efficiency improvements may be considered a conservative assumption and may result in even higher potentials for future electricity savings.

Once we have determined the types of technology and the extent to which they are expected employed in different countries and sectors, we can start quantifying the electricity savings and associated $CO_2$ and air pollution reductions expected from several of the technology switches that replace the use of HFCs. Hence, in addition to the direct climate benefits of HFC emission reductions, transitioning away from HFCs can catalyze additional climate benefits through improvements in the energy efficiency of the refrigerators, air conditioners, freezers, and other products and equipment that currently use HFCs.

Historically, refrigerant conversions, driven by refrigerant phase-outs under the MP, have catalyzed significant improvements in the energy efficiency of refrigeration and A/C systems—up to 60% in some subsectors (Zaelke et al., 2013). Similar improvements are expected under an HFC phase-down following the KA targets. For example, recent demonstration projects for utilizing low-GWP alternatives to HFCs presented by the Climate and Clean Air Coalition (CCAC) calculated energy savings of 15-30% and carbon footprint reductions of 60-85% for refrigeration in commercial food stores (Borgford-Parnell

et al., 2015; UNEP/CCAC, 2016). According to three research studies completed in Brazil, inverter units using lower GWP refrigerants can save up to 67% energy compared to fixed speed units with high GWP R-410A (UNEP/TEAP, 2019). Energy-related emissions can be reduced with lowered cooling demands, more efficient equipment, and operating strategies that maximize system performance (Calm, 2006; Mills, 2011; Sharma et al., 2014; Shah et al., 2015; Purohit et al., 2016; Dreyfus et al., 2017; Sharma et al., 2017; Zaelke and Borgford-Parnell, 2015; IEA, 2018; Purohit et al., 2018b; Park et al., 2019). Shah

et al. (2013) find that even the best currently available technology offers large efficiency improvement opportunities (35-70% reduction in energy consumption from the market average) in room air-conditioners. The current cost-effective efficiency improvements range from 20% to 30% reduction in energy consumption based on a consumer perspective. Based on their operating profiles, even small efficiency improvements translate into significant reductions in GHG emissions (Phadke et al., 2014). Goetzler et al. (2016) estimated 73–76% of global $CO_2$eq emissions from air-conditioning systems in 2010 to be indirect

emissions from the energy use. Recent estimates based on scientific assessments of ozone depletion indicate that improvements in energy efficiency in refrigeration and air-conditioner equipment during the KA transition to low-GWP alternative refrigerants can potentially double the climate benefits of the HFC phase-down (WMO, 2018). In addition to energy efficiency improvements from technical adjustments of the cooling equipment (viz. stationary air-conditioning, commercial, domestic, industrial and transport refrigerators), there is also expected to be a small potential for energy efficiency improvement from

the transition of high-GWP into low-GWP HFCs for given cooling equipment (Schwarz et al., 2011; Barrault et al., 2018; Shah et al., 2019). Both these sources of energy efficiency improvements are considered in this study, while only the latter were considered in Purohit and Höglund-Isaksson (2017) and Höglund-Isaksson et al. (2017).

For the purpose of this study, information on expected energy efficiency improvement potentials through technical adjustments in stationary cooling equipment, were provided by the Lawrence Berkeley National Laboratory (See: Table S2 of the SI). Two





different sets of assumptions were provided; a "technical" and an "economic" energy efficiency potential. The former reflects an efficiency improvement potential considered technically possible, i.e., representing an upper limit for expected energy efficiency improvements, while the latter reflects an efficiency improvement considered economically profitable and represents a minimum energy efficiency improvement. No similar information on expected energy efficiency improvements in mobile air conditioning (MAC) was provided and such improvements have therefore not been considered in this study. Note

that while building design and urban planning can significantly reduce heating or cooling load[7] (IEA, 2013), such options were not considered in this study as the focus here is on energy efficiency enhancements due to uptake of alternative cooling technologies to replace HFCs. Note also that the technical losses of electricity in transmission and distribution (T&D) segments have been taken into account (Brander et al., 2011) whereas non-technical losses (NTL)[8] e.g. due to theft, have not been considered in the estimation of electricity saving potentials. Table S2 of the SI provides information on the unit energy

consumption (UEC) of stationary cooling technologies identified by LBNL and how these have been interpreted in this study in terms of energy efficiency improvement potentials in different sectors/technologies when moving from a pre-KA baseline to low-GWP alternative scenarios.

Lower electricity consumption translates into reduced emissions of $CO_2$, air pollutants such as $SO_2$, $NO_X$ and $PM_{2.5}$, and short-lived climate pollutants (SLCPs) such as black and organic carbon (BC/OC) and methane ($CH_4$). While reductions in

greenhouse gas emissions add to climate change mitigation, co-benefits in the form of reduced air pollution translate into health and ecosystem improvements (Nemet et al., 2010; Markandya et al., 2018; Vandyck et al., 2018). Commercial and residential buildings are known to use more electricity on hotter days (Schaeffer et al., 2012; Valor et al., 2001). The electricity generation units that respond to this increased demand are major contributors to sulfur dioxide ($SO_2$) and nitrogen oxides (NOx), both of which have direct impacts on public health, and contribute to the formation of secondary pollutants including

ozone and fine particulate matter. Abel et al. (2017) found a 3.9% increase in electricity generation per °C that was consistent with Sailor (2001) 0.4−5.3% per °C sensitivity range. Further, NOx emissions sensitivity of 3.60 ± 0.49% per °C (Abel et al., 2017) was consistent with He et al. (2013) range of 2.5−4.0% per °C using similar methodology and region but a different time period.

The GAINS model contains a database on region-specific emission factors for a range of air pollutants and greenhouse gases

from energy production and consumption (IIASA-GAINS, 2019). From this source, we take implied emission factors per GWh electricity consumed for each pollutant listed above and in reflection of expected country- and year- specific fuel mixes used

---

[7] The building envelope determines the amount of energy needed to heat and cool a building, and hence needs to be optimized to keep heating and cooling loads to a minimum. A high-performance building envelope in a cold climate requires just 20-30% of the energy required to heat the current average building in the Organisation of Economic Co-operation and Development (OECD). In hot climates, the energy savings potential from reduced energy needs for cooling are estimated at between 10% and 40% (Dreyfus et al., 2017).

[8] Technical losses occur naturally due to power dissipation in transmission lines, transformers etc. Electricity theft forms a major chunk of the NTL that includes illegal tapping of electricity from the feeder, bypassing the energy meter, tampering with the energy meter and several physical methods to evade payment to the utility company (Depuru et al., 2011).



in power plants in the IEA-WEO 2017 Current Policies Scenario[9] (CPS), New Policies Scenario[10] (NPS) and Sustainable Development Scenario[11] (SDS), respectively, in the timeframe to 2040 (see: Figure S2 of the SI). While the implied emission factors for all other pollutants but $CH_4$ reflect country- and year- specific emissions from combustion of fuels in the power

sector, upstream $CH_4$ emissions from extraction and transmission of fossil fuels used in the power sector are only assessed at the global level due to a lack of information about future fossil fuel trade flows. Hence, the implied emission factors for $CH_4$ reflect global year-specific factors consistent with the weighted average of upstream $CH_4$ emissions embedded in an average unit of electricity consumed. Note that the SDS represents a low carbon scenario consistent with a 2 °C (i.e., 450 ppm) global warming target for this century, and with considerably lower air pollution due to a high degree of replacement of fossil fuel

use with renewable energy. Detailed implied emission factors are available from IIASA's GAINS model only in the timeframe to 2050. The country-specific implied emission factors for air pollutants per GWh electricity consumed representative for year 2050 have therefore been kept constant over the entire period 2050 to 2100.

    In conclusion, Table 1 summarizes the 31 different scenarios generated and analyzed in this study. As outlined in Section 2.1, there are three pre-KA baseline scenarios: Baseline –SSP1, Baseline –SSP3, and a *Cooling for All* baseline. The Baseline –

SSP3 and the *Cooling for All* baseline have been used as starting points for four alternative HFC reduction scenarios; a Kigali Amendment (KA) scenario, a KA high Energy Efficiency (KA-EE) scenario, a Maximum Technically Feasible Reduction (MTFR) scenario, and a MTFR high Energy Efficiency (MTFR-EE) scenario. The high Energy Efficiency scenarios are specified for the "technical" and "economic" energy efficiency improvement potentials described above. For each of the four HFC reduction scenarios with energy efficiency improvements, global and regional estimates of expected electricity savings

and associated impacts on $CO_2$ emissions, air pollutants and SLCPs have been estimated assuming compliance with the KA targets and under maximum technically feasible reductions (MTFR). Finally, for each high EE scenario, three variants of implied emission factors for GHGs ($CO_2$ and $CH_4$) and air pollutants have been used reflecting the three IEA-WEO 2017 energy scenarios, namely, the CPS, NPS and SDS. In this way, the future air pollution projections span a range of possible future energy sector developments.

The KA scenarios (KA and KA-EE) have been developed to analyze the implications of achieving the HFC phase-down targets set out in the KA and specified for four different country/party groups. For each group, the relative HFC phase-down targets differ due to different baselines, HFC consumption freeze years and HFC phase-down schedules (see: Section S2 of the SI). The sector-specific mitigation strategy identified for each of the four KA party groups is determined by the respective marginal

---

[9] This scenario only considers the impact of those policies and measures that are firmly enshrined in legislation as of mid-2017. It provides a cautious assessment of where momentum from existing policies might lead the energy sector in the absence of any other impetus from government.

[10] The NPS aims to provide a sense of where today's policy ambitions seem likely to take the energy sector. It incorporates not just the policies and measures that governments around the world have already put in place, but also the likely effects of announced policies, including the Nationally Determined Contributions (NDCs) made for the Paris Agreement (PA).

[11] This scenario outlines an integrated approach to achieving internationally agreed objectives on climate change, air quality and universal access to modern energy. Further information is available at IEA-WEO (2017) and Rafaj et al. (2018).



abatement cost curves (Höglund Isaksson et al., 2017). Savings on electricity costs make up an important part of the abatement
cost. Because of the larger potentials for energy efficiency improvements assumed here compared with Höglund-Isaksson et
al. (2017), marginal abatement costs are generally lower in this study. Accordingly, a revised set of marginal abatement cost
curves have been generated for all HFC sectors, by each party group, and for each five-year interval to understand the expected
technology compositions after countries have taken action to meet the KA targets. The MTFR scenarios have been developed
to assess the maximum technically feasible reduction of HFCs at the sectoral and regional levels when not considering cost
constraints, but assuming the same sets of energy efficiency improvements as outlined in the KA-EE scenarios. The abatement
potentials in both the KA and MTFR scenarios reflect reductions in emissions through the application of technologies that are
currently commercially available and already tested and implemented, at least to a limited extent. Apart from improvements
in energy efficiency, there is no further improvement assumed in terms of technology removal efficiency of HFCs (which is
anyway complete or close to complete in most sectors) or in terms of investment and non-energy related operation and
maintenance costs.

### 3 Alternatives to high-GWP HFCs

To avoid the use and emissions of both HFCs and HCFCs, a variety of climate-friendly, energy efficient, safe and proven
alternatives are available today (UNEP, 2011; CCAC, 2019). In fact, for most applications where HFCs and HCFCs are still
used in the world, more climate friendly alternatives can be used. However, due to different thermodynamic and safety
properties of the alternatives, there is no "one size fits all" solution applicable to all equipment categories. The suitability of a
certain alternative must be evaluated for each category of product and equipment and also taking account of the level of ambient
temperature at the location where the product and equipment is being used and other factors such as safety codes and
flammability ratings (Abdelaziz et al., 2016; Purohit et al., 2018a).
In recent years, there has been a focus on natural refrigerants (pressurized $CO_2$, hydrocarbons, and ammonia), low-GWP HFCs,
as well as on unsaturated HFCs (also known as HFOs) used alone or in blends with saturated HFCs to replace fluids with high-
GWP. A recent increased use of hydrocarbons (e.g., iso-butane and propane), ammonia, and pressurized carbon dioxide is
expected to continue into the future. Many of these alternatives are widely used in non-Article 5 countries in response to
national or regional regulations that require reductions in HFC use. Many of these technologies are starting to become available
in Article 5 countries and the level of availability is rapidly increasing.
Table 2 lists alternatives that are currently used on a commercial scale and considered in the GAINS model for assessing
mitigation potentials. Moreover, the model considers good practice measures: leakage control during use and recovery of the
refrigerant after end-of-life of the equipment.



# 4 Results and Discussion

## 4.1 HFC emissions

Pre-KA baseline HFC/HCFC emissions consistent with the macroeconomic development of the IEA-WEO 2017 in the period 2005-2040 and with the SSP3 in the period 2050-2100, are presented in Figure 1. For historical years 2005, 2010 and 2015, the contribution from HFC emissions to global greenhouse gas emissions are estimated at 0.46, 0.73 and 1.1 Gt $CO_2$eq, respectively, with an additional 0.27, 0.40 and 0.23 Gt $CO_2$eq release of HCFCs in the respective years. In 2010, 22% of HFC emissions are released from stationary air conditioning, 15% as HFC-134a from mobile air conditioners, 31% from

commercial, industrial, transport and domestic refrigeration, 18% as HFC-23 emissions from HCFC-22 production for emissive and feedstock use, and 14% from use in aerosols, foams, solvents, fire-extinguishers and ground-source heat pumps. Between 2005 and 2050, pre-KA baseline HFC emissions are estimated to increase by a factor of nine, as shown in Figure 1. The growth is mainly driven by a twelve-fold increase in demand for refrigeration and air-conditioning services, which in turn is driven by an expected increase in per capita wealth in developing countries combined with the effect of replacing CFCs and

HCFCs with HFCs in accordance with the revised MP preceding the KA. Under the revised MP, HCFCs in emissive use should be virtually phased out by 2030, but still allowing for servicing of the existing stock until 2040. Baseline HFC emissions are expected to increase to 4.3 Gt $CO_2$eq in 2050 and to 6.2 Gt $CO_2$eq in 2100. The slower increase in the second half of the century is due to saturation in many markets. The expected pre-KA baseline HFC emissions in 2050 are within the range (4.0-5.3 Gt $CO_2$eq) of previous estimates by Velders et al. (2015).

As shown in Figure 2, rapid growth in pre-KA baseline emissions is expected in Article 5 (developing) countries with an approximately eleven-fold increase between 2005 and 2100. China is expected to contribute one-quarter of global HFC emissions in 2050, closely followed by India (21%). Between 2050 and 2100, HFC use in China and India is increasingly saturated and these two countries emit about one third of global HFC emissions in 2100. For the EU-28, pre-KA baseline HFC emissions in 2050 are lower than in 2005 due to implementation of stringent F-gas regulations, whereas corresponding

emissions in the USA increase by a factor of two under existing regulations.

HFC emissions per capita in residential air-conditioning and domestic refrigeration sectors in the SSP3 and *Cooling for All* pre-KA baseline scenarios are presented in Table 3. Due to the increased penetration of room air-conditioners and domestic refrigerators in the *Cooling for All* baseline scenario, HFC emissions per capita in Article 5 parties are 7% and 36% higher in 2050 and 2100, respectively, as compared to the SSP3 baseline scenario.

Figure 1 also presents global pre-KA HFC baseline emissions by key cooling sectors in the three baseline scenarios discussed in Section 2 (including also SSP1). In the *Cooling for All* baseline scenario, HFC emissions could reach 6.8 Gt $CO_2$eq by 2100, driven primarily by an increased cooling demand in the residential sector. As a sensitivity case, HFC emissions in the SSP1 baseline scenario reach 6.1 Gt $CO_2$eq by 2100. Hence, the SSP3 pre-KA baseline emissions fall between the *Cooling for All* and the SSP1 baseline scenarios. In the SSP3 -KA scenario, HFC emissions decline gradually over the analysed period reaching

92% and 95% removal of pre-KA baseline emissions on an annual basis in 2050 and 2100, respectively. Faster emission





reductions than those mandated by the Kigali Amendment represent an additional opportunity for climate change mitigation (Cseh, 2019). The SSP3 -MTFR scenario (lower dashed line) shows that it is considered technically feasible for KA party groups to move earlier in terms of emission reductions and to remove more than 99% of annual emissions in the period 2035 to 2100. Figure S3 presents the HFC/HCFC emissions under the pre-KA baseline and alternative scenarios by different party
groups.

Table 4 presents the corresponding cumulative emissions over the entire period 2018 to 2100. At the global level, cumulative HFC emissions are estimated at 363 Gt $CO_2$eq in the pre-KA SSP3 baseline scenario and at 378 Gt $CO_2$eq in the pre-KA *Cooling for All* baseline scenario. In the sensitivity case using the SSP1 drivers, global cumulative HFC emissions are estimated at 355 Gt $CO_2$eq, which is about 2% less than in the SSP3 baseline scenario. For both the SSP3 and *Cooling for All* baseline
scenarios, stringent compliance with the KA is expected to reduce cumulative HFC emissions by 87% below baseline, whereas maximum technically feasible implementation of abatement technology (MTFR) is expected to reduce cumulative baseline emissions by 96-97%. Over the period 2018-2100, this converts into cumulative HFC emissions of 48 Gt $CO_2$eq when complying with the KA and 13 Gt $CO_2$eq if implementing MTFR. For respective KA party groups, the relative reductions in cumulative emissions 2018-2100 ranges between 84-93% for full compliance with the KA and between 94-99% for full
implementation of MTFR. The lower range values represent party groups with countries that already have legislation implemented to limit the use of HFCs, while the upper range values represent the reduction potential for party groups with countries that currently do not regulate HFC use.

### 4.2 Cost curves

Figure 3 shows the estimated marginal abatement cost curves for global HFC emission reductions under technical and
economic energy efficiency potentials in 2030, 2050 and 2100, respectively. The curves describe the marginal abatement cost paths between the pre-KA baseline and the MTFR emission levels. The red circles in Figure 3 indicate the respective points at the cost curves where the KA targets are being met. For Article 5 countries, there are low cost or even negative (i.e., net profitable) cost options available to meet the KA targets until 2030 due to large potentials to improve on the energy efficiency in cooling technologies. The more ambitious KA targets for 2050 and 2100 are, however, expected to come at a positive
marginal cost and would accordingly require implementation of additional policy incentives. The marginal abatement cost for achieving the Kigali targets is relatively high for non-Article 5 countries in 2030 due to low cost options already adopted in response to the F-gas regulations already implemented at the regional and national levels in many developed countries. The abatement potential extends over time, primarily due to the expected increase in pre-KA baseline emissions but also due to a gradual phase-in of alternative technology in the short run as technical and economic barriers prevent an immediate full uptake
of available technology. Net savings on abatement costs are primarily expected from replacing HFCs with NH3 in industrial refrigeration, switching from HFCs to propane (HC-290) in residential air conditioning, substituting HFCs for isobutane (HC-600a) in domestic refrigerators, replacing HFCs with hydrocarbons (HC-290) in vending machines, using pressurized $CO_2$ in


remote and integral display cabinets in commercial refrigeration, switching from HFCs to $CO_2$-based systems in transport refrigeration, and switching from high to low-GWP HFCs (e.g., HFC-152a) or $CO_2$-based systems in foam blowing.

### 4.3 Co-benefits due to HFC phase-down with enhanced energy efficiency

#### 4.3.1 Electricity savings

Figure 4 presents the technical and economic electricity saving (TWh) potentials when moving from the pre-KA baselines (SSP3 and *Cooling for All*) to corresponding alternative scenarios (KA and MTFR). Globally, the annual technical and economic electricity saving potentials under the KA are estimated at 7882 and 4821 TWh, respectively, in 2050 relative the SSP3 baseline scenario. The annual electricity saving potentials almost double in absolute terms by 2100 as compared to 2050. In the *Cooling for All* scenario the annual technical electricity saving potential are slightly higher than in the SSP3, reaching 8169 TWh in 2050 and 15595 TWh in 2100. The annual technical and economic electricity saving potentials in the alternative scenarios (KA and MTFR) by different party groups are illustrated in Figure S4 and Table S3 of the SI. Note that in the MTFR scenarios, the estimated technical potential is slightly smaller than in the KA. The reason is that the KA scenario is constructed assuming uptake of technologies (to meet the KA reduction targets) in the order of increasing marginal cost for HFC replacement. Options at the very high end of the marginal abatement cost curve (e.g., pressurized $CO_2$) have slightly lower warming potentials than hydrocarbons and HFOs, but also use more electricity (Groll and Kim, 2007; Astrain et al., 2019). It is accordingly an effect of technology switches at the very high end of the marginal abatement cost curve for HFC removal, e.g., hydrocarbons and HFOs replaced by pressurized $CO_2$.

For illustrative purposes, Figure 5 displays a comparison of future annual electricity savings to the total global consumption of electricity as estimated for years 2050 and 2100 in the AIM/CGE SSP3 baseline scenario (Riahi et al., 2017). As shown, if the full technical potential to improve energy efficiency in cooling is implemented as part of efforts to comply with the KA targets, the electricity savings would make up 26% and 22% of expected global electricity consumption in 2050 and 2100, respectively. If only the economic potential to improve energy efficiency in cooling is implemented, the corresponding savings would make up 15% and 13% of expected global electricity consumption in 2050 and 2100, respectively. Hence, the future electricity saving potentials in the cooling sector are significant.

#### 4.3.2 Impacts on greenhouse gas emissions

The electricity-savings presented in Figure 4 can be converted to approximate reductions in GHG ($CO_2$ and $CH_4$) emissions from electricity generation if we combine them with implied emission factors for $CO_2$ and $CH_4$ that reflect the expected country- and year- specific fuel mixes used in power plants in the IEA-WEO 2017 CPS, NPS and SDS, respectively (see Figure S2 of the SI). Figure 6 presents GHG emission reductions in the alternative (KA and MTFR) scenarios due to electricity savings induced by HFC phase-down and under full implementation of technical and economic energy efficiency improvements, respectively, as well as for a range of implied emission factors deriving from the CPS, NPS and SDS,



respectively. The corresponding GHG emission reductions by different KA party groups are presented in Figures S5-S6 of the SI.

Compliance with the KA and full realization of the technical energy efficiency improvement potentials, mean annual reductions in global $CO_2$ emissions estimated at 1.4 Gt in 2050 and 4.4 Gt in 2100 relative a pre-KA SSP3 baseline scenario and using GAINS implied emission factors derived for the IEA-WEO 2017 CPS scenario. For the same set of assumptions, annual global methane ($CH_4$) emissions from extraction of fossil fuels used in the power plant sector are estimated lower by 9 and 15 Mt $CH_4$ in 2050 and 2100, respectively, relative the pre-KA SSP3 baseline scenario. This corresponds to about two percent of expected business-as-usual $CH_4$ emissions from global anthropogenic sources in 2050 (Höglund-Isaksson et al., 2020). As expected, the corresponding annual $CO_2$ mitigation relative the *Cooling for All* baseline is slightly larger at 1.5 Gt in 2050 and 5.1 Gt in 2100 than for the SSP3 baseline. The estimated reductions in $CO_2$ and $CH_4$ emissions from electricity savings are lower when using implied emission factors derived for the IEA-WEO17 NPS and SDS energy sector scenarios than for the CPS, because of higher penetrations of clean fuels (gas, renewables etc.) and uptake of energy efficiency measures in the power sector.

Converting $CH_4$ reductions into $CO_2$eq units using GWPs over 100 years without climate-carbon feedback effects (IPCC, 2013) and adding these to the $CO_2$ reductions allow for calculating total reductions in greenhouse gas emissions due to electricity savings when reducing HFCs to comply with the KA. These are illustrated in Figure 6 for the technical (panel a) and economic (panel b) potentials for energy efficiency improvements. Depending on the energy sector development (CPS, NPS or SDS), annual greenhouse gas emission reductions due to realization of the full technical potential for energy efficiency improvements in cooling are assessed at between 0.8 and 3 Gt $CO_2$eq in 2050 and between almost 2 and 5.5 Gt $CO_2$eq in 2100. Greenhouse gas savings when realizing the economic potential for electricity savings are estimated at about half of that.

We can convert the reduction in HFC emissions into $CO_2$eq terms using GWPs over 100 years without climate-carbon feedback effects (IPCC, 2013). Adding these GHG reductions to those from electricity savings, give us an estimate of total reductions in greenhouse gas emissions due to a phase-down of HFCs. These are shown in Figure 7 with GHG reductions relative a pre-KA SSP3 baseline shown in Panal a) and relative a pre-KA *Cooling for All* baseline shown in Panel b). Results are presented for all the variants of future energy sector development pathways considered (i.e., CPS, NPS and SDS). Compared to a pre-KA baseline, meeting the KA means total annual GHG emissions being lower by between 4.8 and 7.3 Gt $CO_2$eq in year 2050 and between 7.3 and 12.1 Gt $CO_2$eq in 2100. Table 5 presents the cumulative reductions in overall GHG emissions due to both HFC phase-down and a realization of potential energy efficiency improvements. Results are presented by KA party groups and global. Compliance with the KA targets mean avoiding 631 Gt $CO_2$eq of greenhouse gas emissions between 2018 and 2100. About 58% of this cumulative reduction can be attributed to the substitution of HFCs with other low-GWP alternatives, while about 42% can be attributed to electricity savings that derive from the realization of the technical potential to improve energy efficiency in cooling equipment.





### 4.3.3 Impacts on air pollution

Other potentially important environmental benefits of reduced demand for electricity in cooling are reduced air pollution and related adverse effects on human health and ecosystems. Figure 8 presents reductions in air pollutant emissions due to electricity savings associated with the alternative (KA and MTFR) scenarios. The upper set of graphs (panels a-e) show
emission reductions when technical energy efficiency improvement potentials are fully implemented, while the bottom set of graphs (panels f-j) show the corresponding impacts when economic energy efficiency improvement potentials are fully implemented. In 2015, space cooling was responsible for 9% of global emissions of Sulphur dioxide ($SO_2$) emissions from the power sector and 8% of nitrogen oxides (NOx) and fine particulate matter ($PM_{2.5}$) emissions from the power sector (IEA, 2018). Our results indicate that meeting the KA targets means global $SO_2$ emissions in the power sector are reduced by 10%
and 12% relative the SSP3 and *Cooling for All* baselines, respectively, and when assuming implied emission factors from the CPS development of the energy sector (Figure 8a). For the same set of assumptions, annual global NOx emissions in the power sector are expected 16% lower than baseline emissions in 2050 (Figure 8b), while global $PM_{2.5}$ emissions from the power sector are 8% and 9% lower than in the SSP3 and *Cooling for All* baselines, respectively (Figure 8c). Due to a higher penetration of clean fuels in the power sector, reductions in all air pollutant emissions are more limited in the NPS and SDS as compared
to the CPS energy sector development.

Considering the limited contribution of the power sector to total global emissions of these air pollutants, the overall impact on global air pollutant emissions is relatively small at less than 4% according to information on total global emissions in 2050 taken from the GAINS model (IIASA-GAINS, 2019) for the same energy sector development. This small impact makes it difficult to quantify any potential health and ecosystems impacts in a meaningful way.

In addition to air pollutants, we have also assessed the impacts on black and organic carbon (BC/OC) from enhanced energy efficiency in cooling technology. BC/OC are short-lived climate forcers (SLCFs) and as such contribute to global warming. The atmospheric fate and climate impacts of black carbon (BC) from different regions could differ considerably (Berntsen et al., 2006; Reddy and Boucher, 2007). The net effects of BC and organic carbon (OC) on temperature and precipitation are potentially significant, especially at regional scales, because BC and OC have relatively short atmospheric lifetimes (days to
week). These features mean BC/OC are not well mixed in the atmosphere (Bond et al., 2004; Hansen and Nazarenko, 2004; Forster et al., 2007) and therefore not possible to relatively easily convert into $CO_2eq$ terms using GWPs like for other GHGs that are well mixed due to long atmospheric lifetimes. Therefore, we present results for BC/OC impacts in Figure 8 instead of together with the impacts on GHGs in Figure 7. The results indicate that meeting the KA targets means global BC emissions from the power sector are 4% lower in 2030 and 6% lower in 2050 relative the baseline scenarios (Figure 8d). Similarly, global
OC emissions from power plants are 13% lower in 2050 relative the baseline scenarios (Figure 8e). Considering that the power plant sector accounts for less than 0.5% of global BC and OC emissions from all sources (IIASA-GAINS, 2019), the global impact on these emissions from a HFC phase-down is likely minimal.


**5 Conclusions and Policy Recommendations**

Hydrofluorocarbons (HFCs) are widely used as cooling agents in refrigeration and air conditioning, as solvents in certain

industrial processes, for foam blowing and as aerosol propellants. Emissions of HFCs are strong greenhouse gases and as such targeted for reduction to mitigate climate change. The Kigali Amendment (KA) to the Montreal Protocol (MP) from 2016 sets out phase-down pathways to 2050 for the worldwide use of HFCs. Users are encouraged to transition to alternative agents with low global warming potentials (low-GWP). Enhancement of energy efficiency as part of such a transition is a strategic, near-term opportunity to reap significant additional climate and clean air benefits. This study presents long-term scenarios of HFC

emissions to the year 2100 under varying degrees of stringency in climate policy and assesses potential co-benefits in the form of electricity savings and associated reductions in greenhouse gas (GHG) and air pollutant emissions through improved energy efficiency in stationary cooling. The following inferences can be drawn based on this study:

- Prior to the commitments made under the KA, baseline emissions of HFCs are expected to increase from about 0.5 to 4.3 Gt $CO_2$eq between 2005 and 2050, reaching between 6.2 and 6.8 Gt $CO_2$eq in 2100, depending on whether or

not all households in hot climatic conditions install residential air conditioning. The growth is mainly driven by an eighteen-fold increase in demand for refrigeration and air conditioning services, which in turn is driven by an expected increase in per capita wealth in developing countries, a warmer future climate, combined with the effect of replacing CFCs and HCFCs with HFCs in accordance with the 2007 revision of the Montreal Protocol. Cumulative HFC emissions over the entire period 2018 to 2100 are estimated at 363 and 378 Gt $CO_2$eq in respective baseline scenarios.

This is a considerable share of the entire future budget of less than 800 Gt $CO_2$eq that IPCC (2018) estimates as available for the world to remain well below 2 °C warming above the pre-industrial level.

- Full compliance with the commitments made by parties to the KA through replacement of HFCs with low-GWP alternatives, means cumulative HFC emissions of less than 50 Gt $CO_2$eq between 2018 and 2100. With maximum technically feasible implementation of existing control technology and without the delays in implementation built

into the KA, cumulative HFC emissions could be as low as 13 Gt $CO_2$eq between 2018 and 2100, thereby removing about 97% of cumulative pre-KA baseline emissions.

- If carefully addressed during the transition to low-GWP alternatives, improvement potentials for energy efficiency in cooling technologies are extensive and can bring significant electricity savings. We estimate compliance with the KA When fully implementing the technical potential for energy efficiency improvements in cooling, we estimate that

compliance with the KA can bring electricity savings that correspond to more than 20% of the world's entire future electricity consumption. With the energy efficiency improvements limited to the economically profitable applications, electricity savings in cooling could still make up as much as 15% of future electricity consumption.

- Compliance with the KA means avoiding 631 Gt $CO_2$eq of greenhouse gas emissions between 2018 and 2100. About 58% of this cumulative reduction can be attributed to the substitution of HFCs with other low-GWP alternatives,

while about 42% can be attributed to electricity savings that derive from the realization of the technical potential to



improve energy efficiency in cooling equipment. Hence, significant additional reductions in global warming can be achieved if policy-makers address energy efficiency improvements in cooling technology simultaneously with requirements to substitute the use of HFCs with low-GWP alternatives.

- Electricity savings also mean reduced air pollutant emissions from the power sector with related positive effects on human health and ecosystems. We estimate that meeting the KA targets while also implementing the full technical potential for energy efficiency improvements in cooling technologies, can lower future global $SO_2$ emissions from the power sector by up to 10%-12%. Corresponding future impacts on NOx emissions are 16% lower emissions in the power sector relative the baselines and 8-9% lower $PM_{2.5}$ emissions. Considering that the power sector accounts for a smaller share of global emissions of $SO_2$, $NO_x$ and $PM_{2.5}$, the overall impact of electricity savings in cooling on global air pollutant emissions is less than 4%. The impact on black carbon (BC) and organic carbon (OC) emissions is even smaller, 6% and 13% lower emissions from the power sector in 2050. Considering that the power sector contributes less than 0.5% to global BC and OC emissions from all anthropogenic sources, the impact on these emissions from electricity savings in cooling are negligible.

A key policy finding is the importance of paying careful attention to the electricity-savings that can be reaped in the transition away from HFCs in stationary cooling appliances, as the associated greenhouse gas reductions are significant.

*Competing interests.* The authors declare that they have no conflict of interest.

*Acknowledgements.* Financial support from the ClimateWorks Foundation (grant number #IIA-18-1452) is gratefully acknowledged. The authors would like to thank Gabrielle Dreyfus from the Institute for Governance & Sustainable Development (IGSD) and Jyoti Prasad Painuly from UNEP/DTU Partnership for their helpful and constructive comments on an initial version of the manuscript. We would also like to thank Adriana Gomez-Sanabria for research assistance with deriving implied emission factors for electricity consumption.



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





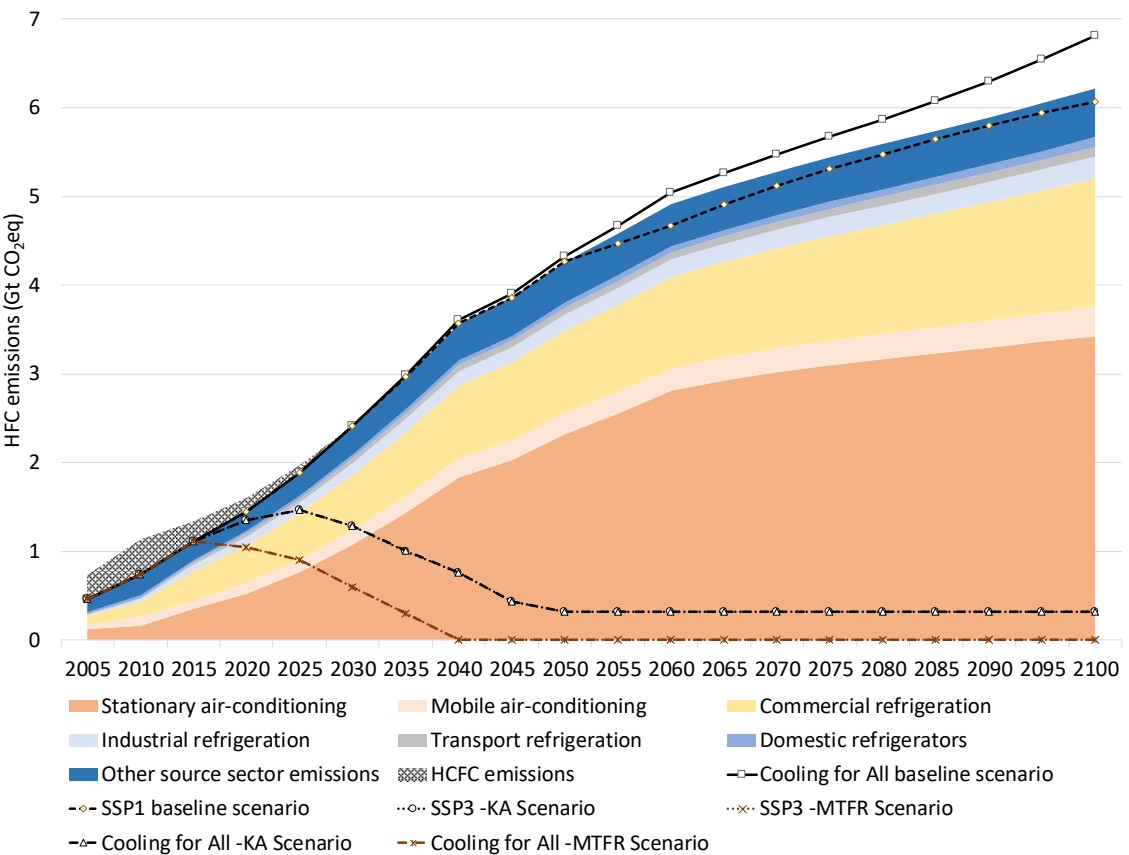

**Figure 1: Pre-KA SSP3 baseline HFC emissions (with baseline SSP1 and *Cooling for All* shown for comparison) and respective alternative scenarios (KA/MTFR). Note that *Cooling for All* -KA and *Cooling for All* -MTFR scenarios are not visible due to the small differences in mitigation scenarios to SSP3 -KA and SSP3 -MTFR.**





**Figure 2: Pre-KA SSP3 baseline HFC emissions by regions**



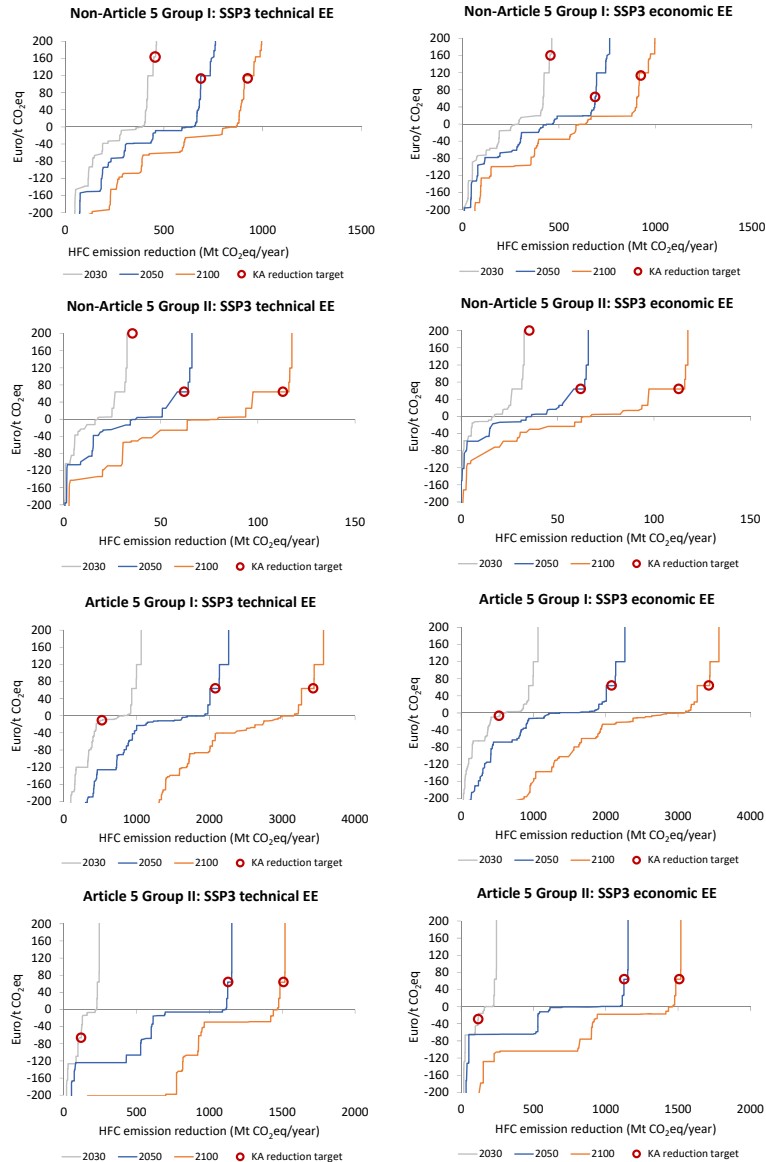

**Figure 3: Marginal abatement cost curves (starting from a pre-KA SSP3 baseline consistent with the IEA-WEO17 New Policies scenario) to reduce HFC emissions by KA party groups and indicating the KA HFC reduction targets in 2030, 2050 and 2100. Left- and right- side panels represent marginal abatement cost curves assuming "technical" and "economic" energy efficiency improvements, respectively.**





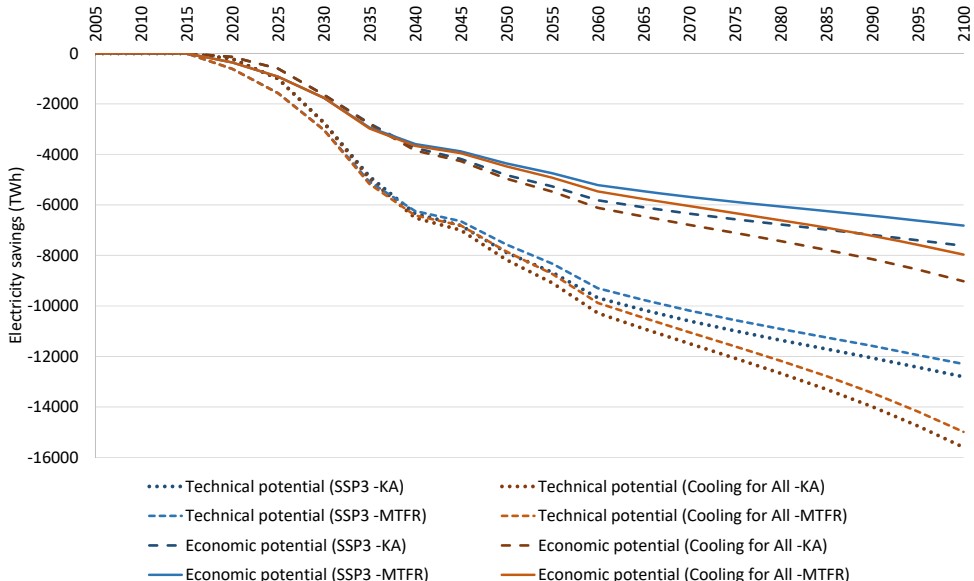

**Figure 4: Technical and economic electricity saving (TWh) potentials in HFC reduction scenarios (KA and MTFR) relative pre-KA baselines (SSP3 and *Cooling for All*)**






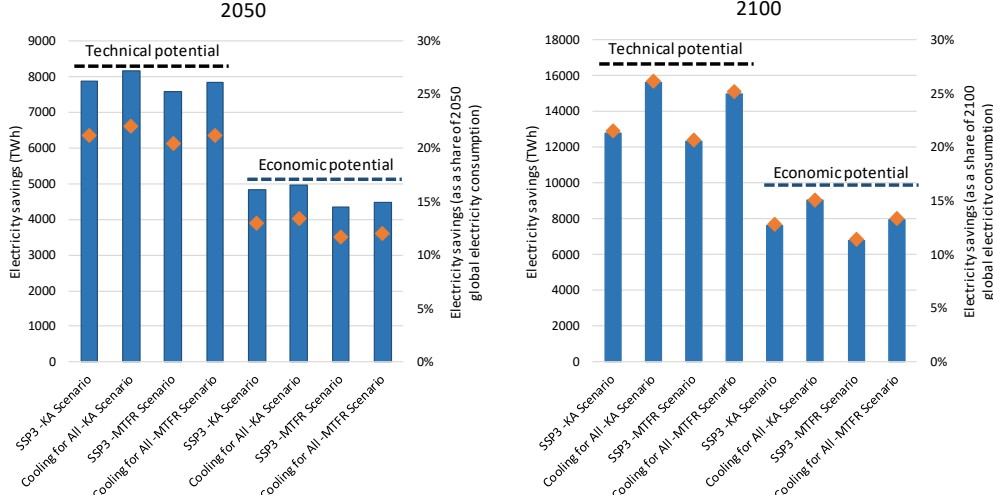

**Figure 5: Annual electricity saving potentials when moving from pre-KA baselines (SSP3 and *Cooling for All*) to HFC reduction scenarios (KA and MTFR), in absolute TWh (blue bars) and as a fraction of expected future global electricity consumption in the AIM/CGE SSP3 baseline scenario (Riahi et al., 2017) (orange dots).**






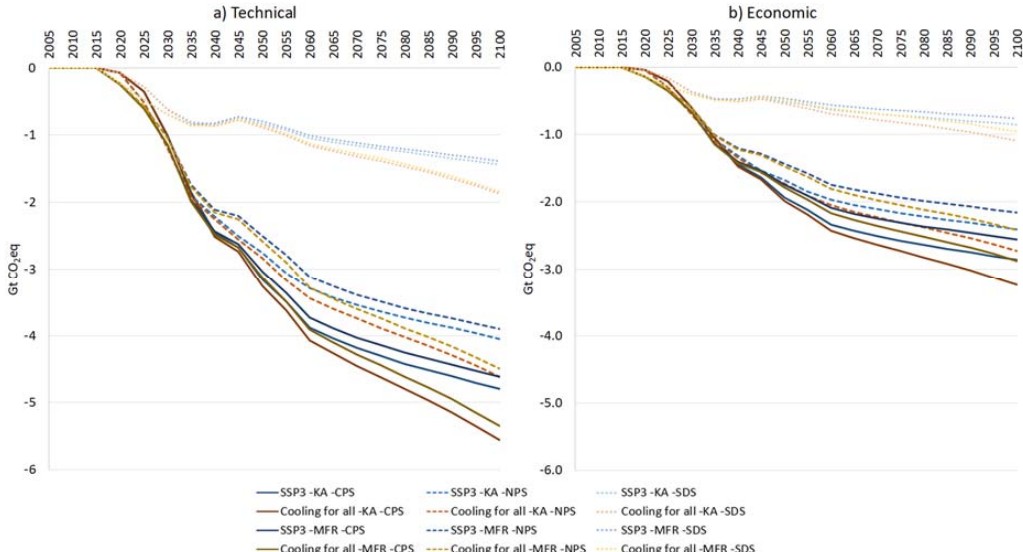

**Figure 6: Annual GHG emission reductions from electricity savings in the KA and MTFR scenarios relative the pre-KA baseline scenarios (SSP3 and *Cooling for All*). Results for technical energy efficiency improvements are shown in Panel a) and for economic energy efficiency improvements in Panel b).**






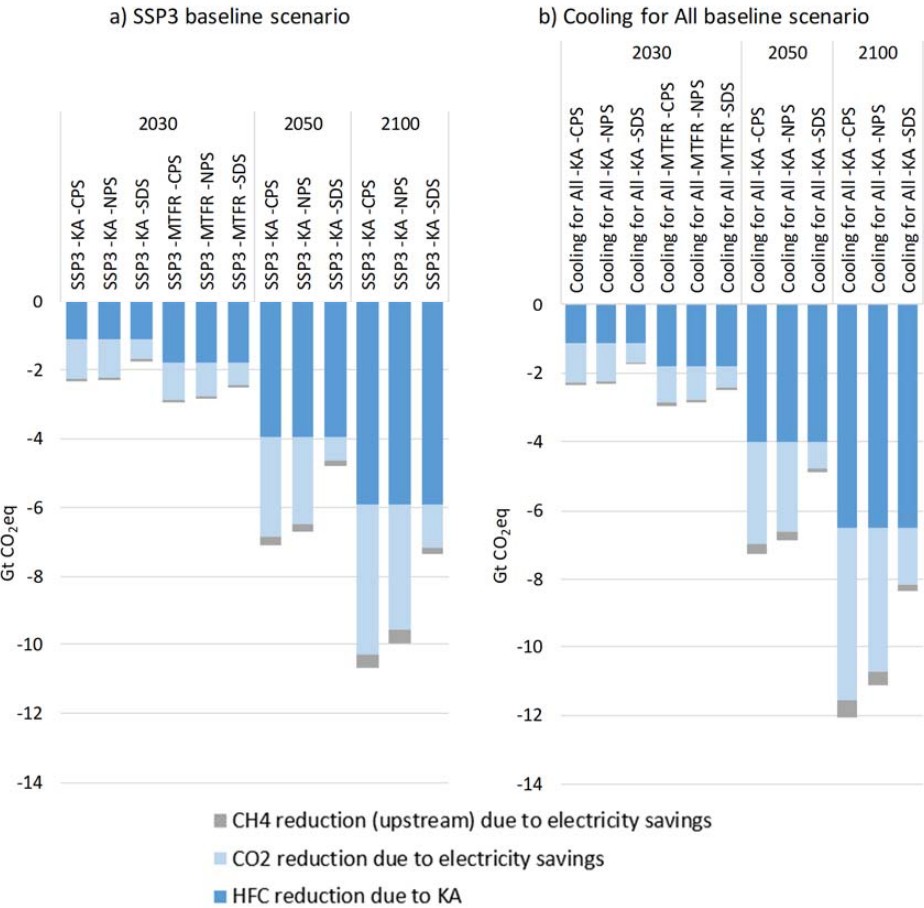

**Figure 7: GHG mitigation (in Gt CO₂eq) due to enhanced energy efficiency benefits under KA in the alternative scenarios with respect to the a) SSP3 baseline scenario and b) Cooling for All baseline scenario. In 2050 and 2100 differences between KA and MTFR scenarios are negligible.**






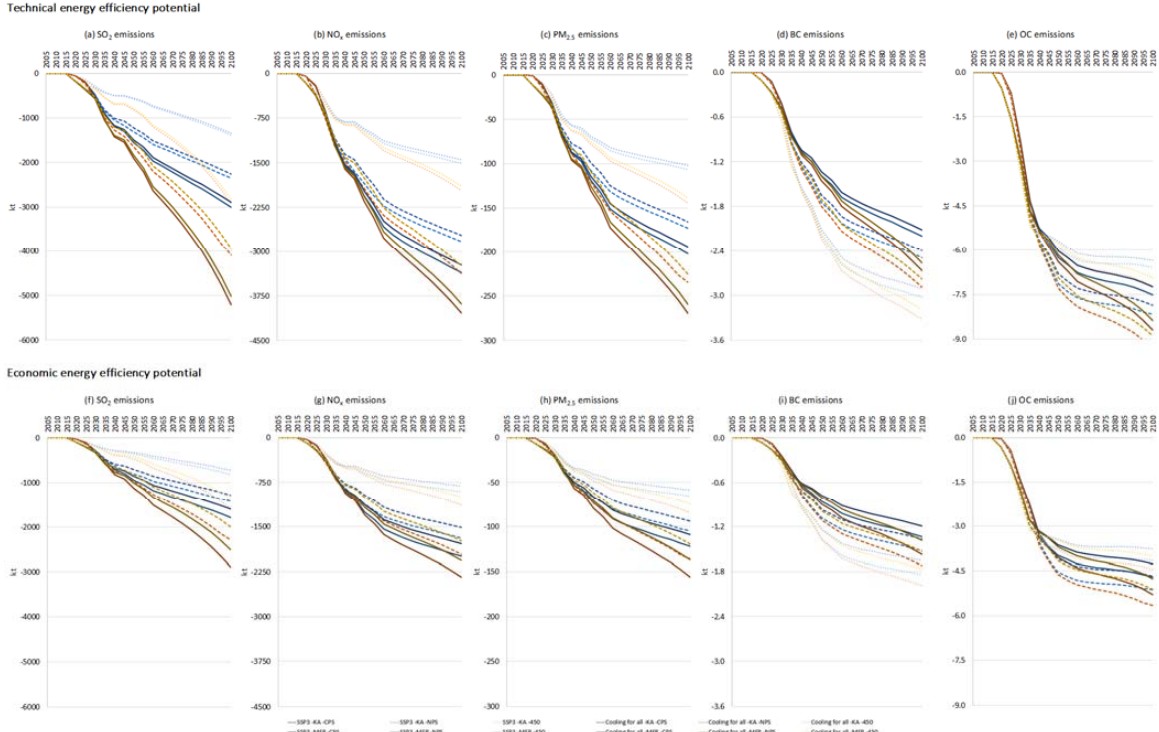

**Figure 8: Impacts on air pollutant and SLCP emissions due to electricity savings associated with alternative HFC phase-down paths.**




**Table 1: Overview of the 31 variants of HFC/co-benefits emission scenarios analysed in this study** (CPS refers to the Current Policies Scenario, NPS to New Policies Scenario, and SDS to Sustainable Development scenario of IEA-WEO 2017).

| | Baseline | a) Kigali Amendment (*KA*) | b) KA with High Energy Efficiency (*KA-EE*) | | | | | | c) Maximum Technically Feasible Reduction (*MTFR*) | d) MTFR with High Energy Efficiency (*MTFR-EE*) | | | | | |
|---|---|---|---|---|---|---|---|---|---|---|---|---|---|---|---|
| | | | Technical EE potential | | | Economic EE potential | | | | Technical EE potential | | | Economic EE potential | | |
| | | | CPS | NPS | SDS | CPS | NPS | SDS | | CPS | NPS | SDS | CPS | NPS | SDS |
| Baseline –SSP1 | X | | | | | | | | | | | | | | |
| Baseline –SSP3 | X | X | X | X | X | X | X | X | X | X | X | X | X | X | X |
| *Cooling for All* | X | X | X | X | X | X | X | X | X | X | X | X | X | X | X |





**Table 2. Sector specific Low-GWP options considered in GAINS**

| Sector | Low-GWP Alternatives |
|---|---|
| Aerosol | ALT_HC (e.g. HC-290), ALT_HFO (e.g. HFO-1234ze), ALT_HFC (e.g. HFC-152a) |
| Commercial refrigeration | ALT_HC (e.g. HC-290), ALT_HFO (e.g. HFO-1234yf), ALT_CO$_2$, ALT_HFC (e.g. HFC-32) |
| Domestic refrigerators | ALT_HC (e.g. HC-600a) |
| Fire-extinguishers | FK (e.g. FK-5-1-12) |
| Foam | ALT_HC (e.g. iso-pentane), ALT_HFO (e.g. HFO-1234ze), ALT_HFC (e.g. HFC-152a), ALT_CO$_2$ |
| Ground source heat pumps | ALT_HC (e.g. HC-290), ALT_HFO (e.g. HFO-1234yf), ALT_CO$_2$, ALT_HFC (e.g. HFC-32) |
| Industrial refrigeration | ALT_NH$_3$, ALT_CO$_2$ |
| Mobile air-conditioning | ALT_HFO (e.g. HFO-1234yf), ALT_CO$_2$ |
| Solvents[*] | Iso-paraffin/siloxane (KC-6) |
| Stationary air-conditioning | ALT_HC (e.g. HC-290), ALT_HFO (e.g. HFO-1234yf), ALT_HFC (e.g. HFC-32), ALT_CO$_2$ |
| Transport refrigeration | ALT_HC (e.g. HC-290, HC-1270), ALT_CO$_2$, ALT_HFC (e.g. HFC-32) |

[*]GAINS also consider a complete ban on HFC based solvents as a control option.





**Table 3: Pre-KA baseline HFC emissions per capita in residential air-conditioning and domestic refrigerators under the SSP3 and *Cooling for All* scenarios**

| Party group | Scenario | Sector | HFC emissions per capita (kg/capita) | |
|---|---|---|---|---|
| | | | **2050** | **2100** |
| Article 5 | SSP3 baseline scenario | Room air-conditioners | 107.9 | 144.3 |
| | | Domestic refrigerators | 5.9 | 9.0 |
| | *Cooling for All* baseline scenario | Room air-conditioners | 114.9 | 196.7 |
| | | Domestic refrigerators | 6.7 | 9.4 |
| Non-Article 5 | SSP3 baseline scenario | Room air-conditioners | 88.6 | 139.8 |
| | | Domestic refrigerators | 3.9 | 5.3 |
| | *Cooling for All* baseline scenario | Room air-conditioners | 88.6 | 139.8 |
| | | Domestic refrigerators | 3.9 | 5.3 |




**Table 4: Cumulative HFC emissions in the pre-KA Baseline and corresponding KA and MTFR scenarios by KA party groups and over the entire period 2018 to 2100.**

| Scenarios | Cumulative HFC emissions (Gt $CO_2$eq) | | | | |
|---|---|---|---|---|---|
| | Non-Art.5 (Group–I) | Non-Art.5 (Group–II) | Art. 5 (Group–I) | Art. 5 (Group–II) | Global |
| pre-KA SSP3 baseline | 66.8 | 6.1 | 199.7 | 90.6 | 363.2 |
| ● *Under KA compliance* | 10.5 | 0.6 | 30.5 | 6.6 | 48.2 |
| ● *Under MTFR* | 4.2 | 0.2 | 7.2 | 1.3 | 12.9 |
| pre-KA *Cooling for All* baseline | 66.8 | 6.2 | 212.7 | 91.9 | 377.5 |
| ● *Under KA compliance* | 10.5 | 0.6 | 30.6 | 6.6 | 48.2 |
| ● *Under MTFR* | 4.2 | 0.2 | 7.3 | 1.3 | 13.0 |
| pre-KA SSP1 baseline | 75.5 | 5.7 | 197.3 | 76.8 | 355.3 |






**Table 5: Cumulative reductions in GHG emissions 2018-2100 due to electricity-savings induced by HFC phase-down when assuming technical energy efficiency improvement potentials, by Kigali Amendment party groups.**

| Scenarios | GHG reductions due to KA and enhanced energy efficiency (Gt $CO_2eq$) | | | | |
|---|---|---|---|---|---|
| | Non-A5 Group-I | Non-A5 Group-II | A5 Group-I | A5 Group-II | Global |
| **SSP3 -KA –CPS** | 98.8 | 12.2 | 329.5 | 190.4 | 631.0 |
| **SSP3 -KA –NPS** | 95.2 | 6.7 | 299.2 | 183.5 | 584.6 |
| **SSP3 -KA –SDS** | 71.9 | 6.3 | 243.5 | 119.8 | 441.4 |
| *Cooling for All* **-KA –CPS** | 98.8 | 12.4 | 359.3 | 193.9 | 664.4 |
| *Cooling for All* **-KA –NPS** | 95.2 | 6.8 | 324.7 | 186.7 | 613.4 |
| *Cooling for All* **-KA –SDS** | 71.9 | 6.4 | 266.0 | 122.4 | 466.7 |
| **SSP3 -MTFR –CPS** | 97.4 | 11.8 | 327.5 | 188.2 | 625.0 |
| **SSP3 -MTFR –NPS** | 94.0 | 6.1 | 298.0 | 181.6 | 579.6 |
| **SSP3 -MTFR –SDS** | 71.6 | 6.1 | 243.6 | 119.7 | 441.0 |
| *Cooling for All* **-MTFR –CPS** | 97.4 | 6.2 | 356.7 | 191.6 | 651.9 |
| *Cooling for All* **-MTFR –NPS** | 94.0 | 6.2 | 324.2 | 184.9 | 609.2 |
| *Cooling for All* **-MTFR –SDS** | 71.6 | 6.2 | 266.5 | 122.4 | 466.8 |
