# Peer review of "Electricity savings and greenhouse gas emission reductions from global phase-down of hydrofluorocarbons"

_Atmospheric Chemistry and Physics, 2020_

## Referee Comment (RC1) · Anonymous Referee #1 · 2 Apr 2020

The authors discuss the emissions of HFCs under several scenarios, with and without the controls of the Kigali Amendment. The novel aspect is that they not only consider the direct climate effects, but also the indirect effects, through changes in the energy use and related air quality aspects. The paper is scientifically sound and the results are interesting and policy relevant. The presentation of the results, though, needs to be improved. There are too many figures with too many panels and lines, which makes it hard to get the main message. The abstract also needs more focus on what is new and not presenting results that have been shown by others also before. I think the paper is acceptable for publication in ACP, after the presentation has been improved.

Main comments: The abstract needs more focus and needs to be shortened. Focus the abstract on what is new (energy savings and air quality aspect), not on results similar to those that have been presented in papers already before. The results on avoided HFC emissions are presented in the conclusions (section 5) and don't need a prominent place in the abstract. The paper contains too many figures; they distract the reader from the main message. Most figures also contain a lot of lines which makes them hard to read and to get the main message out of them. Figure 3: two panels as an example is enough, the rest can be put in the SI. Figure 4 is not needed, since Figure 5 shows the same information in a much clearer way. Figure 6: is this figure needed, if yes, reduce the number of lines. Figure 7 is good and clear. Figure 9 is not readable, too many panels and too many lines. Replace with one clear figure and move the rest to the SI. Some figures also need larger legend. The paper contains a lot of acronyms which makes it not easy to read. The authors should try to avoid acronyms when they are not needed and mostly spell them out in tables and figures, or at least explain the acronyms in all the captions of figures and tables.

Specifics comments: L10-24: These lines in the abstract could be shortened significantly. Only at L24 new information is presented. In L24-29 I would also mention the effects of the economic vs technical mitigation potential. This is a very important and policy relevant result. L11-12: HFCs are not the primary substitute for ODSs under the Montreal Protocol. In many applications ODSs have been replaced by not-in-kind substitutes, such as in cleaning and foam blowing, while hydrocarbons have been used in large quantities in small refrigeration units. I would write "They have been used in large quantities as..." L33: "...and emissive use'. Maybe better to write "...and use as refrigerant" L40: Spell out HFO when it is first mentioned. L45: Please specify the composition of the party groups in the main text (or footnote or caption). Now it is only specified in the SI. Also, the word 'group' is confusing and Group I and group II even more so. In the Protocol groups are defined in Annexes as a set of chemical species. A suggestion: use A5 group A, B, nonA5 group A, B. L115-119: SSP3 is selected as primary scenario and SSP1 as a sensitivity case. I find the logic not

very convincing. The largest differences between in all SSP scenario (1 to 5) occur after 2040 and more even later, so that fact that SSP3 is closest to the IEA scenario up to 2040 is not a very strong argument. So if you select SSP3 as primary scenario than use the highest and lowest of the SSP scenarios (I guess 4 and 5 in your case) for sensitivity. They show the range of results, especially for the second half of this century. Clearly, under the KA they all collapse on to one curve. L180: What do you mean with 'HFC removal efficiency'? To me it could mean, replacing HFCs with other substances or NIK technologies, but also HFC capture and destruction. L181: Again, what is 'removal of HFCs'? I also don't understand the rest of the sentence. 'removal of HFCs is close to complete. . .not affect conclusions regarding the HFC phase-down'. If removal is complete does that mean the phase down is complete? Please clarify this sentence. L218-219: 'no information . . . was provided . . .' This is an odd argument. Improvements in MAC are clearly taking place, although maybe not directly related to energy efficiency. How it will effect $CO_2$ emissions and air quality is a completely different study and I can understand that that is the reason it is not taken into account here. I would rephrase the sentence. L232-233: 'The electricity generation units. . .'. Please specify what units will be used first. I can imagine that this is different in different countries. What did you assume? L277: What is meant with '. . .at least to a limited extent.' This weakens the rest of the sentence considerably. L289-294: I agree with this paragraph, but it would be good to have a reference for it. L289: Be careful with the term low-GWP alternatives (see my comment with Table 2) L345-347: You have to mention somewhere that in, e.g., the EU, Japan, Australia HFC regulations are already in place and preceded the time the KA came into force. The situation in the US is complicated. L348: Very useful paragraph. The corresponding figure (3) needs to be simplified (see below). Have the national/regional regulations that are already in place been taken into account here? In the EU for example the phasedown of HFCs is already well underway. L436-440: There are many acronyms in section 4.3.3. Please spell out CPS, NPS, SDS. This makes it easier to read. L783: Figure 3: Simplify this figure by moving panels to the SI. The message comes much better across with only

two panels. L790: Figure 4 shows negative numbers for savings. I think this is confusing. Or 'savings' or negative/reduced electricity use. In also Figure 5 where positive savings are shown. L800: Same as in figure 4. Confusing to show negative emission reductions. L805: Same as figure 4 and 5 L810: Figure 8: Very unclear: too many panels and too small numbers. This figure has to be improved. Table 1: I think the table can be simplified, since almost all scenarios have an 'X'. L815: Table 2: HFC-32 is mentioned here as a low GWP alternative. This is confusing. There has been a lot of discussions in among parties to the Montreal Protocol on the term low-GWP. A value of 150 is sometimes considered 'low' because it is a value used in the EU regulation. HFC-32 is not considered a low GWP alternative. It is used as an alternative with a 'lower' GWP than the compound it replaces. Please use the terms 'alternatives' and 'low-GWP' carefully.

---

## Referee Comment (RC2) · Anonymous Referee #2 · 29 Apr 2020

This is a very nice paper and it is timely. The calculations use the well-established GAINS model and uses various assumptions, most of which are documented. The paper shows that the use of low GWP substitutes (including non-fluorinated refrigerants) for the high-GWP HFCs along with efficiency gains in better equipment design would help reduce climate change. This occurs through the reduction in the lower greenhouse effect of the substitutes and lesser CO2 emission from lower electricity usage.

The main concern I have about this paper is: Is ACP the right venue for this paper that is mostly about economic analyses and non-atmospheric assumptions. I have debated this for a few days and came to the conclusion that it would not hurt atmospheric scientists to read this paper to understand factors that go into decision making and the level of knowledge about the atmosphere that is used in such decision making! It should be eye-opening to them. I will leave it up to the Editor to make this call on suitability. But, I stand on the side of publishing it here!

I have a number of comments for the authors to consider, some are small and some are more important. I list them below.

Very major comment:
Personally, I don't think that there should be policy recommendations. I would cast the same recommendations as options and the gains made from such options. Policy recommendations do not go too well in science papers!

Major comments:
1. The future warming is not the same across the globe. There are major regional and latitudinal differences. Also, the mean temperature is not what determines the use of cooling. It is the changes in the high temperatures. Do you account for these factors in your analysis? If you do not, you should explicitly state it and point out the uncertainties that you get from such an assumption.
2. I actually agree with your choice of baseline. But, you need to discuss at least briefly how much difference it will make going forward. We are already in 2020!
3. How sensitive are your calculations to the assumption the efficiency gains made from switching from CFCs/HCFCs to HFCs is translated to going from high-GWP HFCs to lower GWP substitutes? Is there an upper limit to the efficiency gains that can be achieved? Does this efficiency gain take into the change in the thermodynamic efficiency loss due to higher temperatures (not the global mean, but the location dependent predicted high temperatures)? Can this efficiency be improved if particular attention is paid to this factor? It would be nice to see something discussed here.
4. Can you make some comments about the gains made if renewables were used? Afterall, you are projecting to 2100!

Minor comments:
1. Not all HFCs are very potent greenhouse gases. You need to qualify your statements.

2. Your quoted GWP is for a mix of HFCs. You need to state this. Also, I think you are using 100 year GWPs, which are not necessarily appropriate since most HFCs have much shorter lifetimes and hence their shorter horizon GWPs are larger. How does that affect the near term gains/disbenefits?
3. Somewhere in your model you have a specific fuel mix used to generate electricity. It would be useful to explicitly state those.
4. I am impressed with your citation list! You are very comprehensive!
5. Have you considered that aerosols offset GHG of $CO_2$? This happens only up to a point and then it does not. This influence can have major influences in the future. (See Murphy and Ravishankara, PNAS, 2018).
6. I am sorry to say that your figures are not easy to read, especially if somebody is partially colorblind. The lines are impossible to see, the axes are rather poorly formatted and too numerous to see. I assume (hope) that you will improve all your figures.

---

## Author Comment (AC1) · 17 Jul 2020

Referee 1 (Anonymous)

The authors discuss the emissions of HFCs under several scenarios, with and without the controls of the Kigali Amendment. The novel aspect is that they not only consider the direct climate effects, but also the indirect effects, through changes in the energy use and related air quality aspects. The paper is scientifically sound, and the results are interesting and policy relevant. The presentation of the results, though, needs to be improved. There are too many figures with too many panels and lines, which makes it hard to get the main message. The abstract also needs more focus on what is new and

not presenting results that have been shown by others also before. I think the paper is acceptable for publication in ACP, after the presentation has been improved.

Authors' Response: We thank the Anonymous Referee for his/her constructive comments and many helpful suggestions on how to improve the manuscript. Below we provide detailed point by point replies to the questions. We would like to emphasize that a large amount of additional information on existing policies for phasing down hydrofluorocarbon (HFC) consumption, baseline and HFC phase-down schedule of Article-5 and non- Article-5 Parties and results by different party groups, has been included for paper size reasons in the supplementary material (see the attachment). In the revised version of the manuscript, we have improved the abstract, figures and overall presentation of the results as suggested by the reviewer.

Main comments:

1. The abstract needs more focus and needs to be shortened. Focus the abstract on what is new (energy savings and air quality aspect), not on results similar to those that have been presented in papers already before. The results on avoided HFC emissions are presented in the conclusions (section 5) and don't need a prominent place in the abstract.

Authors' Response: As suggested, we have shortened the abstract (from 355 words to 266 words) in the revised version of the manuscript primarily focusing on co-benefits (electricity savings and reduction in air pollutants and greenhouse gas emissions) of the HFC phase-down under Kigali amendment (KA) to the Montreal Protocol.

2. The paper contains too many figures; they distract the reader from the main message. Most figures also contain a lot of lines which makes them hard to read and to get the main message out of them. Figure 3: two panels as an example is enough, the rest can be put in the SI. Figure 4 is not needed, since Figure 5 shows the same information in a much clearer way. Figure 6: is this figure needed, if yes, reduce the number of lines. Figure 7 is good and clear. Figure 8 is not readable, too many panels

and too many lines. Replace with one clear figure and move the rest to the SI. Some figures also need larger legend.

Authors' Response: As suggested, we have improved the font size and split Figure 3 in two parts – Marginal abatement cost curves (MACCs) starting from a pre-Kigali SSP3 baseline consistent with the IEA-WEO17 New Policies scenario and reducing HFC emissions by KA party groups under a) technical energy efficiency improvements in the revised manuscript; and b) economic energy efficiency improvements in the supplementary section (Figure S4).

In the revised version of the manuscript, Figure 4 on "Technical and economic electricity saving (TWh) potentials in HFC reduction scenarios (KA and MTFR) relative pre-KA baselines (SSP3 and Cooling for All)" is deleted as suggested by the reviewer.

As suggested, we have improved the font size and readability of Figure 6 in the revised version of the manuscript.

Once again, we have improved the font size and split Figure 8 in two parts – a) Impacts on air pollutant emissions due to electricity savings are presented in the revised manuscript whereas the b) Impacts on BC/OC emissions due to electricity savings are presented in the supplementary section (Figure S8).

3. The paper contains a lot of acronyms which makes it not easy to read. The authors should try to avoid acronyms when they are not needed and mostly spell them out in tables and figures, or at least explain the acronyms in all the captions of figures and tables.

Authors' Response: As per the reviewer's comments, we have reduced acronyms to the extent possible in the revised version of the manuscript. In addition, we have spelled out most of the acronyms in all figures and tables of the paper.

Specifics comments:

4. L10-24: These lines in the abstract could be shortened significantly. Only at L24

new information is presented.

Authors' Response: As suggested, we have shortened the abstract (from 355 words to 266 words) in the revised version of the manuscript.

5. In L24-29 I would also mention the effects of the economic vs technical mitigation potential. This is a very important and policy-relevant result.

Authors' Response: Corrected in the revised version of the manuscript (See: L19-25). We have rephrased the text and highlighted the effects of the economic vs technical mitigation potential:

"If technical energy efficiency improvements are fully implemented, the resulting electricity savings could exceed 20

In addition, Table 5 of the revised manuscript also presents cumulative reductions in GHG emissions 2018-2100 due to electricity-savings induced by HFC phase-down when assuming economic energy efficiency improvement potentials, by Kigali Amendment party groups.

6. L11-12: HFCs are not the primary substitute for ODSs under the Montreal Protocol. In many applications, ODSs have been replaced by not-in-kind substitutes, such as in cleaning and foam blowing, while hydrocarbons have been used in large quantities in small refrigeration units. I would write "They have been used in large quantities as: :"

Authors' Response: Comment appreciated. As suggested, we made the following change in the revised version of the manuscript (See: L11-12):

"They have been used in large quantities as the primary substitutes for ozone-depleting substances regulated under the Montreal Protocol (MP)."

7. L33: ": : :and emissive use'. Maybe better to write ": : :and use as refrigerant"

Authors' Response: Comment appreciated. As suggested, we made following change in the revised version of the manuscript (See: L28-29).

"As well, HFC-23 is generated as a by-product of chlorodifluoromethane (HCFC-22) production used in refrigerants and as a chemical feedstock for manufacturing synthetic polymers."

8. L40: Spell out HFO when it is first mentioned.

Authors' Response: Corrected, . . . hydrofluoroolefins or HFOs in short. . . in the revised version of the manuscript (See: L36).

9. L45: Please specify the composition of the party groups in the main text (or footnote or caption). Now it is only specified in the SI. Also, the word 'group' is confusing, and Group I and group II even more so. In the Protocol, groups are defined in Annexes as a set of chemical species. A suggestion: use A5 group A, B, nonA5 group A, B.

Authors' Response: Article 5 and non-Article 5 parties are defined within the Montreal Protocol based on their annual calculated level of consumption of any controlled substance per capita. Those that exceed this level of annual calculated consumption are classified as non-Article 5 and those that do not exceed it as Article 5 parties. For the groups, we have used the classification from UNEP Ozon Action (See: http://www.unep.fr/ozonaction/information/mmcfiles/7880-e-Kigali$_FS05_Baselines_Timetable.pdf$). $We simply write Group 1 and Group 2 in the revised version of the manuscript instead of Group-I and Group-II. As sugge$

"The Montreal Protocol Parties are split into four Kigali Amendment groups: a) Non-Article 5, earlier start - Most non-Article 5 countries; b) Non-A5, later start - Russia, Belarus, Kazakhstan, Tajikistan, Uzbekistan; c) Article 5, Group 1 - Most Article 5 countries; and d) Article 5, Group 2 - Bahrain, India, Iran, Iraq, Kuwait, Oman, Pakistan, Qatar, Saudi Arabia and UAE."

10. L115-119: SSP3 is selected as primary scenario and SSP1 as a sensitivity case. I find the logic not very convincing. The largest differences between in all SSP scenario (1 to 5) occur after 2040 and more even later, so that fact that SSP3 is closest to the IEA scenario up to 2040 is not a very strong argument. So if you select SSP3 as

primary scenario than use the highest and lowest of the SSP scenarios (I guess 4 and 5 in your case) for sensitivity. They show the range of results, especially for the second half of this century. Clearly, under the KA they all collapse on to one curve.

Authors' Response: We agree with the reviewer's comment that under KA all SSP scenarios will collapse on to one curve. We have now tried to better motivate our choice of SSP3 as main baseline scenario by adding the following footnote (11) in L128 (Section 2.1):

"With the exception of SSP5 and as shown in Figure S1 of the SI, SSP1 and SSP3 represent roughly the full range of future population and GDP developments in the SSPs. SSP5 is not considered as a baseline in this study, since the dimension of a continued fossil-fuel intensive future vs a decarbonized future is already integrated in the analysis through the range of country-specific implied emission factors from the CPS vs the SDS scenarios of the IEA-WEO2017. In the period beyond 2040, the country- sector- and fuel specific emission factors derived from these scenarios for the year 2040 are kept constant."

11. L180: What do you mean with 'HFC removal efficiency'? To me it could mean, replacing HFCs with other substances or NIK technologies, but also HFC capture and destruction.

Authors' Response: We agree with the reviewer's comment that the way we have used this expression was confusing. We have replaced this expression everywhere with 'efficiency in reducing the climate impact of cooling when replacing HFC use' to make it clearer what we mean (See: L188, Section 2.2).

12. L181: Again, what is 'removal of HFCs'? I also don't understand the rest of the sentence. 'removal of HFCs is close to complete: : :not affect conclusions regarding the HFC phase-down'. If removal is complete does that mean the phase down is complete? Please clarify this sentence.

Authors' Response: We have tried to rewrite these sentences to hopefully be clearer about what we mean (See: L187-194, Section 2.2):

"Note that for given technology options, potential effects of future technological development on costs and the efficiency in reducing the climate impact of cooling when replacing HFCs, have not been considered here. It would also not have a significant impact on conclusions of this study, since the use of HFCs in cooling can be completely replaced by existing alternative low-GWP measures, and cost are not assessed at the absolute level but for the sole purpose of using MACCs to determine the order of technology uptake. Technological development could also mean even larger potentials for energy efficiency improvements than those considered here as technical and economic potentials. Not considering the possibility of such effects here may be considered a conservative assumption, as it could mean there are potentials for even larger future electricity savings."

13. L218-219: 'no information : : : was provided : : :' This is an odd argument. Improvements in MAC are clearly taking place, although maybe not directly related to energy efficiency. How it will affect CO2 emissions and air quality is a completely different study and I can understand that that is the reason it is not taken into account here. I would rephrase the sentence.

Authors' Response: Comment appreciated. As suggested, we have rephrased the sentence in the revised version of the manuscript (L228-230, Section 2.2) as follows:

"Note that energy efficiency improvements take place also when HFCs are replaced in mobile air conditioners (MAC) (Blumberg et al., 2019). These are however not accounted for here as the drivers for associated emission changes are very different from those in stationary sources and more complex to estimate."

14. L232-233: 'The electricity generation units: : :'. Please specify what units will be used first. I can imagine that this is different in different countries. What did you assume?

Authors' Response: To be clearer about what we mean, we have replaced "units" with "plants" (See: L245-248, Section 2.2). The sentence now reads "The electricity generation plants (e. g. coal, oil and gas fired power plants) that respond to this increased demand are major contributors to SO2 and NOx emissions, both of which have direct impacts on public health, and contribute to the formation of secondary pollutants including ozone and fine particulate matter (PM2.5)."

The assumptions for deriving country-, sector- and fuel- specific implied emission factors from the GAINS model are explained further down in the text (Section 2.2).

15. L277: What is meant with ': : :at least to a limited extent.' This weakens the rest of the sentence considerably.

Authors' Response: To avoid confusion, we have deleted the text ". . . . . .at least to a limited extent" from this sentence.

16. L289-294: I agree with this paragraph, but it would be good to have a reference for it. Authors' Response: As suggested, we have added the following references in this paragraph (See: L301, Section 3): 1. Beshr, M., Aute, V., Sharma, V., Abdelaziz, O., Fricke, B. and Radermacher, R.: A comparative study on the environmental impact of supermarket refrigeration systems using low GWP refrigerants, Int. J. Refrigeration, 56, 154-164, https://doi.org/10.1016/j.ijrefrig.2015.03.025, 2015. 2. McLinden, M.O., Brown, J.S., Brignoli, R., Kazakov, A.F. and Domanski, P.A.: Limited options for low-global warming potential refrigerants, Nature Communications, 8, https://doi.org/10.1038/ncomms14476, 14476, 2017. 3. Heredia-Aricapa, Y., Belman-Flores, J.M., Mota-Babiloni, A., Serrano-Arellano, J. and García-Pabón, J.J.: Overview of low GWP mixtures for the replacement of HFC refrigerants: R134a, R404A and R410A, Int. J. Refrigeration, 111, 13-123, https://doi.org/10.1016/j.ijrefrig.2019.11.012, 2020. 4. UNEP: Lower-GWP Alternatives in Commercial and Transport Refrigeration: An expanded compilation of propane, CO2, ammonia and HFO case studies, United Nations Environment Programme (UNEP), Paris, 2016a.

17. L289: Be careful with the term low-GWP alternatives (see my comment with Table 2)

Authors' Response: Thanks for pointing out the error. We have changed the title of Table 2 to "Sector specific alternative options for high-GWP hydrofluorocarbons considered in the GAINS model".

18. L345-347: You have to mention somewhere that in, e.g., the EU, Japan, Australia HFC regulations are already in place and preceded the time the KA came into force. The situation in the US is complicated.

Authors' Response: Comment appreciated. In the supplementary information (SI) section, we have provided a separate section S1 on "Current legislation on HFC control considered in the Baselines" – highlighting HFC control or phase-down policies at regional and national level in Article 5 and non-Article 5 countries.

As suggested, we have added the following text at the end of Section 4.1 (L356-358) of the revised manuscript:

"In non-Article 5 countries (mainly developed countries), national and regional (e.g. EU) regulations have been implemented to limit the use of high-GWP HFCs through limiting imports, production and exports prior to the Kigali amendment entering into force. More specific information about these regulations is available in Section S1 of the SI."

19. L348: Very useful paragraph. The corresponding figure (3) needs to be simplified (see below). Have the national/regional regulations that are already in place been taken into account here? In the EU for example the phasedown of HFCs is already well underway.

Authors' Response: As suggested, we have improved the font size and split Figure 3 in two parts – Marginal abatement cost curves (MACCs) starting from a pre-Kigali SSP3 baseline consistent with the IEA-WEO17 New Policies scenario and reducing

HFC emissions by KA party groups under a) technical energy efficiency improvements in the revised manuscript; and b) economic energy efficiency improvements in the supplementary section (Figure S4 of the SI).

We have considered the national/regional regulations (e.g. EU F-gas regulations) in the baseline scenarios. More specific information about these regulations is available in Section S1 of the Supplementary Information on - Current legislation on HFC control considered in the Baselines. We have referred this section here (L368, Section 4.2).

20. L436-440: There are many acronyms in section 4.3.3. Please spell out CPS, NPS, SDS. This makes it easier to read.

Authors' Response: We have explained all three scenarios (current policies scenario – CPS; new policies scenario – NPS; and sustainable development scenario- SDS) in Section 2.2 in the revised version, and thereafter refer to them consistently as "CPS, NPS, and SDS energy scenarios" in Section 4.3.3.

21. L783: Figure 3: Simplify this figure by moving panels to the SI. The message comes much better across with only two panels.

Authors' Response: As suggested, we have improved the font size and split Figure 3 in two parts – Marginal abatement cost curves (MACCs) starting from a pre-Kigali SSP3 baseline consistent with the IEA-WEO17 New Policies scenario and reducing HFC emissions by KA party groups under a) technical energy efficiency improvements in the revised manuscript; and b) economic energy efficiency improvements in the supplementary section (Figure S4).

22. L790: Figure 4 shows negative numbers for savings. I think this is confusing. Or 'savings' or negative/reduced electricity use. In also Figure 5 where positive savings are shown.

Authors' Response: We have replaced "savings", which indeed was incorrect if expressed in negative numbers, to "Potentials for changes in annual electricity consumption".

23. L800: Same as in figure 4. Confusing to show negative emission reductions.

Authors' Response: Same here, reductions changed to "Changes in annual GHG emissions". Thanks for pointing this out.

24. L805: Same as figure 4 and 5 L810: Figure 8: Very unclear: too many panels and too small numbers. This figure has to be improved.

Authors' Response: As suggested, we have improved the font size and split Figure 8 in two parts: Impact on a) air pollutant emissions (see: Figure 8 of the revised manuscript), and b) BC/OC emissions (Figure S8 of the supplementary section) due to electricity savings associated with alternative HFC phase-down paths.

25. Table 1: I think the table can be simplified, since almost all scenarios have an 'X'.

Authors' Response: In the revised version, we have used "âIJŞ" for the scenarios analyzed and "X" for the scenario not considered (see: Table 1) in this study.

26. L815: Table 2: HFC-32 is mentioned here as a low GWP alternative. This is confusing. There has been a lot of discussions in among parties to the Montreal Protocol on the term low-GWP. A value of 150 is sometimes considered 'low' because it is a value used in the EU regulation. HFC-32 is not considered a low GWP alternative. It is used as an alternative with a 'lower' GWP than the compound it replaces. Please use the terms 'alternatives' and 'low-GWP' carefully.

Authors' Response: We appreciate the reviewers' comments. As suggested, we have changed the title of Table 2 as "Sector-specific alternative options for high-GWP hydrofluorocarbons considered in the GAINS model".

Please also note the supplement to this comment:
https://www.atmos-chem-phys-discuss.net/acp-2020-193/acp-2020-193-AC1-supplement.pdf

[Figure]

**Figure 1: Pre-Kigali SSP3 baseline HFC emissions (with baseline SSP1 and *Cooling for All* shown for comparison) and respective alternative scenarios (Kigali Amendment -KA and Maximum Technically Feasible Reduction -MTFR). Note that *Cooling for All* -KA and *Cooling for All* -MTFR scenarios are not visible due to the small differences in mitigation scenarios to SSP3 -KA and SSP3 -MTFR.**

**Fig. 1.**

HFC emissions (Pg CO₂eq)

Legend:
- Australia
- Brazil
- China
- EU-28
- India
- Indonesia
- Japan
- South Korea
- Mexico
- Russia
- South Africa
- USA
- Rest of the World

**Figure 2: Pre-Kigali SSP3 baseline HFC emissions by regions**

Fig. 2.

Figure 3: Marginal abatement cost curves (MACCs) starting from a pre-Kigali SSP3 baseline consistent with the IEA-WEO17 New Policies scenario and reducing HFC emissions by Kigali Amendment (KA) party groups under technical energy efficiency improvements and indicating the KA HFC reduction targets in 2030, 2050 and 2100.

**Fig. 3.**

[Figure]

**Figure 4: Annual electricity saving potentials when moving from pre-Kigali baselines (SSP3 and *Cooling for All*) to HFC reduction scenarios (Kigali Amendment -KA and Maximum Technically Feasible Reduction -MTFR), in absolute TWh (blue bars) and as a fraction of expected future global electricity consumption in the AIM/CGE SSP3 baseline scenario (Riahi et al., 2017) (orange dots).**

**Fig. 4.**

[Figure]

**Figure 5: Annual greenhouse gas emission reductions from electricity savings in the Kigali Amendment (KA) and Maximum Technically Feasible Reduction (MTFR) scenarios relative the pre-Kigali baseline scenarios (SSP3 and *Cooling for All*). Results for technical energy efficiency improvements are shown in Panel a) and for economic energy efficiency improvements in Panel b).**

**Fig. 5.**

[Figure]

**Figure 6: Greenhouse gas mitigation (in Pg CO₂eq) due to enhanced energy efficiency benefits under Kigali amendment (KA) in the alternative scenarios with respect to the a) SSP3 baseline scenario and b) *Cooling for All* baseline scenario. In 2050 and 2100 differences between KA and Maximum Technically Feasible Reduction (MTFR) scenarios are negligible.**

Fig. 6.

Range for Baseline HFC
emissions pre-KA

HFC emissions with Kigali
Amendment (KA)

HFC emissions with Maximum
Technically Feasible Reduction
(MTFR)

Range for HFC emission
reductions

Range for sum of HFC reductions
and GHG emission reduction due
to technical energy efficiency
improvement

Range for sum of HFC reductions
and GHG emission reduction due
to economic energy efficiency
improvement

**Figure 7: Full range of HFC emissions and mitigation potential under baselines and Kigali Amendment (KA) and Maximum Technically Feasible Reduction (MTFR) scenarios along with HFC and other greenhouse gas mitigation under technical and economic energy efficiency improvement scenarios analysed in this study.**

**Fig. 7.**

[Figure]

**Figure 8: Impacts on air pollutant emissions due to electricity savings associated with alternative HFC phase-down pathways.**

**Fig. 8.**

[Figure]

**Figure S4: Marginal abatement cost curves (MACCs) starting from a pre-Kigali SSP3 baseline consistent with the IEA-WEO17 New Policies scenario and reducing HFC emissions by Kigali Amendment (KA) party groups under economic energy efficiency improvements and indicating the KA HFC reduction targets in 2030, 2050 and 2100.**

**Fig. 9.**

[Figure]

Figure S8: Impacts on BC/OC emissions due to electricity savings associated with alternative HFC phase-down pathways.

**Fig. 10.**

---

## Author Comment (AC2) · 17 Jul 2020

Referee #2 (Anonymous)

This is a very nice paper and it is timely. The calculations use the well-established GAINS model and uses various assumptions, most of which are documented. The paper shows that the use of low-GWP substitutes (including non-fluorinated refrigerants) for the high-GWP HFCs along with efficiency gains in better equipment design would help reduce climate change. This occurs through the reduction in the lower greenhouse effect of the substitutes and lesser CO2 emission from lower electricity usage.

The main concern I have about this paper is: Is ACP the right venue for this paper that is mostly about economic analyses and non-atmospheric assumptions. I have debated this for a few days and came to the conclusion that it would not hurt atmospheric scientists to read this paper to understand factors that go into decision making and the level of knowledge about the atmosphere that is used in such decision making! It should be eye-opening to them. I will leave it up to the Editor to make this call on suitability. But I stand on the side of publishing it here!

I have a number of comments for the authors to consider, some are small, and some are more important. I list them below.

Authors' Response: We thank the Anonymous Referee for his/her constructive comments and many helpful suggestions on how to improve the manuscript. Below we provide a detailed point by point replies to the questions. We would like to emphasize that a large amount of additional information on existing policies for phasing down HFC consumption, baseline and HFC phase-down schedule of Article-5 and non- Article-5 Parties and results by different party groups, has been included – for the paper size reasons - in the supplementary material (see the attachment).

Main comments:

1. Personally, I don't think that there should be policy recommendations. I would cast the same recommendations as options and the gains made from such options. Policy recommendations do not go too well in science papers!

Authors' Response: Comment appreciated. We have rephrased section 5 as "Conclusions" instead of "Conclusions and Policy Recommendations".

2. The future warming is not the same across the globe. There are major regional and latitudinal differences. Also, the mean temperature is not what determines the use of cooling. It is the changes in the high temperatures. Do you account for these factors in your analysis? If you do not, you should explicitly state it and point out the uncertainties

that you get from such an assumption.

Authors' Response: We agree with the reviewer that the warming varies between global regions. While warming has not been uniform across the planet, the upward trend in the globally averaged temperature shows that more areas are warming than cooling. The influence of warming on cooling demand is much higher in tropical and sub-tropical regions, but other factors such as humidity and building performance also play a role. Therefore, in this study, the extension in demand for cooling services has been generated in consistency with the growth in population and macroeconomic indicators and the expected future increase in national/regional cooling degree days (CDDs) as developed and provided by International Energy Agency (IEA), as discussed in Section 2.1. Implemented at a national/regional level in our analysis, the CDDs increase globally on average by nearly 15% between 2016 and 2050 and 20% between 2016 and 2100 in the SSP3 baseline scenario. We have added the following footnote to provide more clarity on our assumptions in the revised version of the manuscript (See: footnote (10), Section 2.1):

"Cooling degree days (CDD) are country/region specific and measure how much (in degrees), and for how long (in days), outside air temperature was higher than a specific base temperature. For the purposes of this study, CDDs are measured in $^{\circ}$C, standardized to 18$^{\circ}$C, and adopted at a country/regional level in consistency with IEA (2018)."

3. I actually agree with your choice of baseline. But you need to discuss at least briefly how much difference it will make going forward. We are already in 2020!

Authors' Response: In the baseline scenarios (SSP3, Cooling for All and SSP1), we have considered, regional (EU) and national policies/regulations for phasing down HFC emissions (See: Section S1 of the SI). As a result, in industrialized countries, particularly Europe, HFC emissions are in decline due to ambitious national and regional policies to regulate F-gas use. However, a large increase is expected from developing

countries (primarily Article 5 parties) primarily in response to increased demand for cooling services and the phase-out of ozone-depleting substances under the Montreal Protocol.

The amount of energy needed to meet demand for space cooling varies mainly according to the type and efficiency of the equipment used, how it is used and how often it is used, as well as the type and thermal efficiency of buildings. The energy consumption per unit of cooling output of cooling technologies currently on sale around the world varies massively. We have used country/region specific information on unit energy consumption as shown in Table S2 of the SI. For the electricity savings, we consider both the technical and energy efficiency improvement potential of stationary cooling technologies due to systems improvement and transition towards low-GWP refrigerants. In addition, we have used a range of future energy sector developments (Current Policies Scenario, New Policies Scenario, and Sustainable Development Scenario) to assess country/region specific implied emissions factors for GHG and air pollutants to get a clear sense of the range of directions in which today's energy sector policy ambitions could impact GHGs and air pollution emissions from electricity savings.

As a result, full compliance with the Kigali Amendment means avoiding 631 Pg CO2eq of greenhouse gas emissions between 2018 and 2100. As explained in the text (Section 4.3.2), about 58% of this cumulative reduction can be attributed to the substitution of HFCs with other low-GWP alternatives, while about 42% can be attributed to electricity savings that derive from the realization of the technical potential to improve energy efficiency in cooling equipment. Hence, significant additional reductions in global warming can be achieved if the Montreal Protocol Parties address energy efficiency improvements in cooling technology simultaneously with requirements to substitute the use of HFCs with low-GWP alternatives.

4. How sensitive are your calculations to the assumption the efficiency gains made from switching from CFCs/HCFCs to HFCs is translated to going from high-GWP HFCs to lower GWP substitutes?

Authors' Response: Comment appreciated. The efficiency gains calculated are from improvements in the equipment (heat exchangers, compressors, valves etc.) and thus mostly independent of the refrigerant(s) used. The switch to lower GWP substitute refrigerants usually entails an efficiency gain or loss on the order of ~5% which we assume would roughly cancel out when aggregated across product categories. Unfortunately, since the final refrigerant alternatives that will eventually be deployed and their characteristics are still being researched, while our work is based on the refrigerants that are currently available, it is not possible to be more specific than this in the current version of the manuscript.

5. Is there an upper limit to the efficiency gains that can be achieved?

Authors' Response: Yes, this is usually dictated by constraints such as thermodynamics, cost, weight, space and installation constraints if the dominant type of technology continues to be vapor compression systems. Current Best Available Technology is still roughly between 30-70% of the thermodynamically ideal efficiency (varying by the other constraints mentioned), as mentioned in Table S2 of SI.

6. Does this efficiency gain take into the change in the thermodynamic efficiency loss due to higher temperatures (not the global mean, but the location dependent predicted high temperatures)? Can this efficiency be improved if particular attention is paid to this factor? It would be nice to see something discussed here.

Authors' Response: No, however, it is anticipated in the scenarios examined in the paper that any losses of efficiency due to changes in temperature in the future are likely to affect both the baseline and higher efficiency technology roughly equally since most refrigerants decline in efficiency at higher ambient temperatures and thus there is not much to gain in efficiency terms by paying further attention to this factor.

7. Can you make some comments about the gains made if renewables were used? Afterall, you are projecting to 2100!

Authors' Response: Yes, this dimension is taken into account by using implied emission factors from IEA's Current Policies Scenario (CPS) and Sustainable Development Scenario (SDS). We have explained the impacts of replacement of fossil fuel use with renewable energy in Section 2.2 (L264-266). The GAINS model contains a database on region-specific emission factors for a range of air pollutants and greenhouse gases from energy production and consumption. From this source, we take implied emission factors per GWh electricity consumed for $CO_2$, $CH_4$, $SO_2$, NOx, PM2.5 and SLCPs (BC and OC) and in reflection of expected country- and year- specific fuel mixes used in power plants in the IEA-WEO 2017 Current Policies Scenario (CPS), New Policies Scenario (NPS) and Sustainable Development Scenario (SDS), respectively, in the timeframe to 2040 (see: Figure S2 of the SI).

Note that the SDS represents a low carbon scenario consistent with a 2 oC (i.e., 450 ppm) global warming target for this century, and with considerably lower air pollution due to a high degree of replacement of fossil fuel use with renewable energy (solar, wind, biomass, etc.). Detailed implied emission factors are available from IIASA's GAINS model only in the timeframe to 2040. The country-, sector-, and fuel- specific implied emission factors for air pollutants per GWh electricity consumed representative for year 2040 have therefore been kept constant over the entire period 2040 to 2100.

The estimated reductions in $CO_2$ and $CH_4$ emissions from electricity savings are accordingly lower when using implied emission factors derived for the IEA-WEO17 SDS energy sector scenarios than for the CPS, because of higher penetrations of clean fuels (gas, renewables etc.) and uptake of energy efficiency measures in the power sector.

Specifics comments:

8. Not all HFCs are very potent greenhouse gases. You need to qualify your statements.

Authors' Response: As suggested, we have rephrased the sentences: (L12-13) −

"However, many HFCs are potent greenhouse gases......" (L32-33) – "Many HFCs are potent greenhouse gases......"

9. Your quoted GWP is for a mix of HFCs. You need to state this. Also, I think you are using 100-year GWPs, which are not necessarily appropriate since most HFCs have much shorter lifetimes and hence their shorter horizon GWPs are larger. How does that affect the near-term gains/disbenefits?

Authors' Response: Comment appreciated. As already indicated in Section 2.1, L100-101, Blends of HFCs have been decomposed and attributed to respective HFC species. For e.g., HFC-410A (R-410A) a zeotropic mixture (a mixture of liquids that boils at a constant temperature, at a given pressure, without change of composition) of 50% HFC-32 and 50% HFC-125, HFC-407C (R-407C) a zeotropic mixture of 23% HFC-32, 25% HFC-125, and 52% HFC-134a. We agree that the lifetime of most of the HFCs is lower than 100 years except HFC-23 and HFC-236fa, GWP100 is lower than GWP20 (IPCC, 2013).

In the revised version of the manuscript, we have added the following paragraph on why we have chosen to use GWP100 (See: L105-114, Section 2.1): "In this study, we have chosen to follow the convention of the policy community to use IPCC global warming potentials over 100 years (GWP100) without climate-carbon feedback effects to convert the varying atmospheric lifetimes and warming potentials for different HFC species to CO2eq units (IPCC, 2013). This convention has been adopted in negotiations for several international climate agreements, e.g., the Kyoto Protocol, in the draft text of the Paris Agreement (UNFCCC, 2018), the standardized Life Cycle Assessment (LCA)/carbon-foot printing approaches (ISO, 2006) and in media and among the general public for assessing the relative climate impacts of given products or activities (Lynch et al., 2020). Despite there being good reasons for questioning this convention, in particular when analysing the impact of short-lived climate forcers (Cain et al., 2019), we find it well-motivated to apply the standard GWP100 metric here as it facilitates the discussion of results in the policy context. A broader assessment of implications of

results on global warming in the short- and long run could be an interesting topic for future research but is considered out of scope for this paper."

There have been proposals for the UNFCCC to adopt a dual-term greenhouse gas accounting standard: 20-year GWPs alongside the presently accepted 100-year GWPs. It is argued that the advantage of such a change would be to more rapidly reduce short term warming and buy time for CO2 reductions. However, these changes could be counterproductive, and the benefits are overstated. The balance of near-term cooling followed by long-term warming would be even worse for 20-year GWPs, because this would "allow" dodging even more CO2 reductions for every unit amount of reduced short-lived greenhouse gas.

10. Somewhere in your model you have a specific fuel mix used to generate electricity. It would be useful to explicitly state those.

Authors' Response: The GAINS model contains a database on country/region-specific emission factors (specific for 174 countries/regions as used in this study) for a range of air pollutants and greenhouse gases from energy production and consumption. From this source, we take implied emission factors per GWh electricity consumed for each pollutant and in reflection of expected country- and year- specific fuel mixes used in power plants in the IEA-WEO 2017 Current Policies Scenario (CPS), New Policies Scenario (NPS) and Sustainable Development Scenario (SDS), respectively, in the timeframe to 2040 (see: Figure S2 of the SI). Note that the SDS represents a low carbon scenario consistent with a 2 oC (i.e., 450 ppm) global warming target for this century, and with considerably lower air pollution due to a high degree of replacement of fossil fuel use with renewable energy. We have elaborated specific fuel mix used to generate electricity in Section 2.2 (L256-266).

11. I am impressed with your citation list! You are very comprehensive!

Authors' Response: Thanks for encouraging words.

12. Have you considered that aerosols offset GHG of CO2? This happens only up to a point and then it does not. This influence can have major influences in the future (See Murphy and Ravishankara, PNAS, 2018).

Authors' Response: Comment appreciated. However, in this study, we have not considered the offsetting effects of the greenhouse gas and aerosol emissions as the primary focus of this study is to assess co-benefits in the form of electricity savings and associated reductions in greenhouse gas and air pollutant emissions due to the global phase-down of hydrofluorocarbons under the Kigali Amendment to the Montreal Protocol.

13. I am sorry to say that your figures are not easy to read, especially if somebody is partially colorblind. The lines are impossible to see, the axes are rather poorly formatted and too numerous to see. I assume (hope) that you will improve all your figures.

Authors' Response: We apologize for the inconvenience. As suggested, we have improved the font size and split Figure 3 in two parts – Marginal abatement cost curves (MACCs) starting from a pre-Kigali SSP3 baseline consistent with the IEA-WEO17 New Policies scenario and reducing HFC emissions by KA party groups under a) technical energy efficiency improvements in the revised manuscript; and b) economic energy efficiency improvements in the supplementary section (Figure S4).

In the revised version of the manuscript, Figure 4 on "Technical and economic electricity saving (TWh) potentials in HFC reduction scenarios (KA and MTFR) relative pre-KA baselines (SSP3 and Cooling for All)" is deleted as suggested by the reviewer#1. In addition, we have improved the font size and readability of Figure 6 (now Figure 5) in the revised version of the manuscript.

Finally, we have improved the font size and split Figure 8 in two parts – a) Impacts on air pollutant emissions due to electricity savings are presented in the revised manuscript whereas the b) Impacts on BC/OC emissions due to electricity savings are presented in the supplementary section (Figure S8).  

References

[revised manuscript text omitted]

---

## Author Response (AR2)

**Journal: Atmospheric Chemistry and Physics**

**Title: Electricity savings and greenhouse gas emission reductions from global phase-down of hydrofluorocarbons**

**MS No.: acp-2020-193**

**Referee #1 (Anonymous)**

*The authors have significantly improved the presentation and figures of the manuscript.*

Authors' Response: We thank the Anonymous Referee for his/her constructive comments and many helpful suggestions on how to improve the manuscript. Below we provide detailed point by point replies to the questions.

**Minor comments:**

1. *L21-23: Mention here that this is under the assumption that the current technologies are used to generate electricity. If the world rapidly switched to sustainable energy the savings are different.*
   Authors' Response: We do agree with the reviewer's comment that the savings in greenhouse gas emissions will be lower if the world rapidly switched to sustainable energy. In L21-23, the higher number - 631 Pg $CO_2$ equivalent (under technical energy efficiency potential) follows the assumption that the current technologies are used to generate electricity under IEA/WEO current policies scenario. The lower number - 411 Pg $CO_2$ equivalent (under economic energy efficiency potential) follows the IEA/WEO sustainable development scenario that outlines an integrated approach to achieving internationally agreed objectives on climate change, air quality and universal access to modern energy. In addition, Table 5 presents the full range of the cumulative reductions in greenhouse gas emissions 2018-2100 under different scenarios analyzed in this study due to electricity-savings induced by HFC phase-down when assuming technical and economic energy efficiency improvement potentials (by Kigali Amendment party groups) following the current policies, new policies and sustainable development scenarios. As suggested, we have rephrased the sentence as follows (L21-24):

   "The combined effect of HFC phase-down, energy efficiency improvement of the stationary cooling technologies and future changes in the electricity generation fuel mix would prevent between 411 and 631 Pg $CO_2$ equivalent of GHG emissions between 2018 and 2100, thereby making a significant contribution towards keeping the global temperature rise below 2°C."

2. *L32: mention "for a 100-yr time horizon"*
   Authors' Response: As suggested, we have rephrased the sentence as follows (L32-34):

"Many HFCs are potent greenhouse gases (GHGs) with a global warming potential (GWP) up to 12400 times that of $CO_2$ per mass unit (IPCC, 2013) *over a 100-year time horizon*."

3.  *L161: "reduced greenhouse gas emissions …"*
    Authors' Response: As suggested, we have rephrased the sentence as follows (L160-163):
    "Extended refrigeration of food would also mean reduced food losses, which apart from having important implications for meeting nutritional needs, would also contribute to *reduced greenhouse gas emissions* from food production and better use of the 23–24% of global cropland and fertilizers currently used to produce food that is eventually wasted (Kummu et al., 2012; Hiç et al., 2016)."

4.  *L303: the term 'non-Article 5 countries' is used here for the first time. Mention in brackets 'developed countries'.*
    Authors' Response: Thanks for pointing this out. As suggested, we have rephrased the sentence as follows (L302-305):
    "Many of these alternatives are widely used in non-Article 5 *(developed countries)* countries in response to national or regional regulations that require reductions in HFC use. The availability and uptake *of these alternatives* is rapidly increasing also in Article 5 countries (Reese, 2018; UNEP, 2019)."

5.  *L464: 'to be used'*
    Authors' Response: As suggested, we have rephrased the sentence as follows (L464-465):
    "Hydrofluorocarbons (HFCs) are manufactured *to be used* as substitutes for ozone-depleting substances that are being phased out globally under Montreal Protocol regulations."

6.  *L465: HFCs are greenhouse gases, not their emissions. Rephrase "HFCs are strong greenhouse gases and as such their emissions are targeted …"*
    Authors' Response: As suggested, we have rephrased the sentence as follows (L465-466):
    "HFCs are strong greenhouse gases, with a global warming effect up to 12,400 times greater than carbon dioxide, and their emissions are rising strongly."

7.  *L498: I don't think it is only up to policymakers to address energy efficiency improvements. Industry and consumers also play a role here.*
    Authors' Response: Thank you for pointing this out. As suggested, we have rephrased the sentence as follows (L498-501):
    "Hence, significant additional reductions in global warming can be achieved if *policymakers, manufacturers, industry and other stakeholders (e.g. consumers, utilities etc.)* address energy efficiency improvements in cooling technology simultaneously with requirements for HFCs substitution."

8. *L501-502: Mention here that this is under the assumption that the current technologies are used to generate electricity. If the world rapidly switched to sustainable energy the savings are different.*
   Authors' Response: As suggested, we have added the following sentence here (L507-509):
   "It may be noted that the higher range follows the assumption that the current technologies are used to generate electricity under the current policies scenario whereas lower range reflect transition towards sustainable energy under the sustainable development scenario."

9. *L509: "as the associated greenhouse gas emission reductions …"*
   Authors' Response: As suggested, we have rephrased the sentence as follows (L512-513):
   "A key policy finding is the importance of paying careful attention to the electricity-savings that can be reaped in the transition away from HFCs in stationary cooling appliances, *as the associated greenhouse gas emission reductions are significant*."

10. *L851: "Greenhouse gas emission mitigation …"*
    Authors' Response: Corrected in the revised version of the manuscript (see: L855 o the revised manuscript).